# SINQ: Sinkhorn-Normalized Quantization
# for Calibration-Free Low-Precision LLM Weights

**Lorenz K. Muller** [1] **Philippe Bich** [1] **Jiawei Zhuang** [1] **Ahmet Celik** [1] **Luca Benfenati** [1] **Lukas Cavigelli** [1]

## Abstract

Post-training quantization has emerged as the most widely used strategy for deploying large language models at low precision. Still, current methods show perplexity degradation at bit-widths $\leq 4$, partly because representing outliers causes precision issues in parameters that share the same scales as these outliers. This problem is especially pronounced for calibration-free, uniform quantization methods. We introduce SINQ to augment existing post-training quantizers with an additional second-axis scale factor and a fast Sinkhorn–Knopp–style algorithm that finds scales to normalize per-row and per-column variances. We show that this approximates activation-aware quantization by recovering column scales from the weight matrix structure that are predictive of the typical activation magnitudes the matrix received during training. Our method has no interactions between layers and can be trivially applied to new architectures to quantize any linear layer. We evaluate our method on the Qwen3 model family, among others. SINQ reduces the perplexity gap on WikiText2 and C4 by over $50\%$ against uncalibrated uniform quantization baselines, incurs zero to negligible compute overhead, and can be further enhanced by combining it with calibration and non-uniform quantization levels. Code is available in this repository.

## 1. Introduction

Post-training quantization (PTQ) is a powerful approach to reducing the cost of neural network inference. Weight quantization reduces the storage, memory, and data movement required to run a neural network. As such, it is useful on its own whenever any of these components bottleneck an inference system's performance. Potential speed-ups of pure weight-quantization are substantial: For example, moving from float16 to int4 weights yields a potential speedup of 4x in memory-bound scenarios. Weight-only quantization is especially popular in LLM deployment because accelerator memory capacity and data movement are often the initial performance bottlenecks in this scenario.

In this paper, we demonstrate that a carefully chosen uncalibrated, uniform quantizer can approach the end-to-end output quality of calibrated quantizers or non-uniform formats while being simpler: Calibration (and even more so end-to-end optimization) is an intuitive approach to improving the output quality of quantized models, but comes with the inherent downsides of possible bias and overfitting (Lin et al., 2024b) and additional compute time required at quantization time (which is prohibitive in continual learning settings). Similarly, non-uniform formats can offer an improvement over integer quantization (Dettmers et al., 2023), but require potentially costly look-ups during inference and cannot be combined with activation quantization in compute-limited scenarios. In brief, if uncalibrated uniform quantization were to reach the same output quality, it would be preferable for these reasons. This paper largely closes the gap between these different approaches to quantization.

The key contributions of this paper are:

- We find that column-wise weight matrix standard deviations are predictive of input scales, allowing for calibration-free pseudo-activation-aware quantization.

- We propose adding a scaling factor along the second axis of to-be-quantized matrix tiles alongside a fast algorithm based on Sinkhorn-Knopp iterations that finds column-wise scales that mimic activation-awareness, while avoiding row-wise kurtosis increase (Sec. 2).

- In numerous experiments across different model scales, we show that our method substantially improves over state-of-the-art baselines for calibration-free quantization methods (Sec. 3).

- We provide code for easy quantization of LLMs using linear layers.

[1]Computing Systems Lab, Huawei Technologies, Switzerland. Correspondence to: Lorenz K. Muller <lorenz.mueller@huawei.com>.

*Proceedings of the 43rd International Conference on Machine Learning*, Seoul, South Korea. PMLR 306, 2026. Copyright 2026 by the author(s).

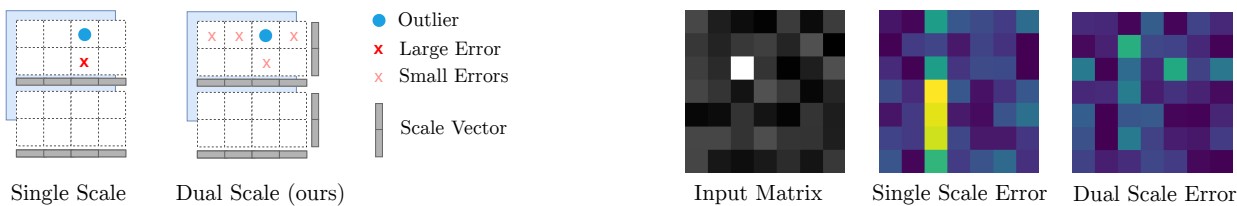

*Figure 1.* If we have scales along both dimensions of a matrix that is to be quantized, we can trade off the impact of outliers between rows and columns, which is impossible in single-scale quantization. Left: Conceptual illustration of error distributions with single or dual-scaling. Right: Example on small matrix.

## 2. Methods

We divide our method into two parts: Firstly, the quantized parameterization, i.e. the mathematical expression used to map between the full precision and the quantized matrix. All quantization methods used in practice, have some set of auxiliary parameters to use in this mapping. Secondly, the representation space, i.e. the space in which we instantiate the full precision matrix when quantizing it.

Throughout we use the pytorch notation, where a linear layer is $\vec{x} \cdot \mathbf{W}^\mathsf{T}$ for input $\vec{x}$ and weight $\mathbf{W}$.

### 2.1. Quantized Parametrization

Typically, one does not simply replace the weight matrix with, for example, an INT4 matrix, but rather divides it into tiles and assigns some higher-precision auxiliary parameters to each tile. Here, we describe different possibilities for the type of auxiliary parameters to use and how to tile the matrix.

#### 2.1.1. PARAMETERIZATION PER TILE

**Scales + Shifts** The most widely used approach uses a scale and a shift vector (e.g., (Badri & Shaji, 2023)), like so:

$$\mathbf{W}_{\text{approx}} = \vec{s} \odot (\mathbf{Q} + \vec{z}) \tag{1}$$

where $\mathbf{W}_{\text{approx}}$ is a $N \times M$ matrix (or matrix tile), $\vec{s}$ is a $N \times 1$ vector, $\vec{z}$ is a $N \times 1$ vector and $\mathbf{Q}$ is a quantized $N \times M$ matrix. Also, the transpose of this with $1 \times M$ vectors is commonly used. All commonly used methods currently use this approach; specifically, all methods we compare in this paper do so.

**Dual-Scales** Instead of supplying a single vector of scales along one dimension of the matrix, we supply two vectors, one along each dimension. Formulaically, we propose:

$$\mathbf{W}_{\text{approx}} = \vec{s} \odot \mathbf{Q} \odot \vec{t} \tag{2}$$

where $\vec{s}$ is a $N \times 1$ vector, $\vec{t}$ is a $1 \times M$ vector and the rest is as above.

The key benefit of Eq. 2 can be illustrated as follows: Say $W_{ij}$ is an outlying large value. By scaling up $s_i$ and scaling down $t_j$ we can trade off quantization errors that will occur in row $i$ for errors in column $j$. See Fig. 1 for an illustration.

**Dual-Scales + Shifts** If we do not mind the potential additional overhead (or rather, if an accuracy improvement justifies it), we can also add shifts to the dual scales:

$$\mathbf{W}_{\text{approx}} = \vec{s} \odot (\mathbf{Q} + \vec{z}) \odot \vec{t} \tag{3}$$

#### 2.1.2. TILING

Typically, tiling for quantization is implemented along one dimension of the matrix that is to be quantized (e.g., HQQ (Badri & Shaji, 2023), AWQ (Lin et al., 2024b)). Consequently, these tiles have rectangular shapes; e.g., a $N \times M$ matrix tiled with tile size $T$ would yield tiles of shape $N \times T$. This could cause a problem with the dual-scale parameterization. Namely, the standard parameterization has $2 \times N \times M/T$ scale and shift parameters, while the dual-scaled only has $N \times M/T + M$. Because of this, we use dual-scale parameterization together with a shift (as in Eq. 3). This has a small additional overhead compared to single-scale + shift parameterization; the total auxiliary parameters are $2 \times N \times M/T + M$. We report total memory use in all our experiments to confirm that this overhead of $M$ additional parameters is negligible.

### 2.2. Representation Space

Before assigning values to the parameters from which we will reconstruct our matrix, we may want to transform the space in which the matrix is represented, to make the reconstruction better aligned with some quality metric (like weight MSE or end-to-end accuracy on some validation data). The two most common among such transformations of the weight space are rotations (like the Hadamard transform (Ashkboos et al., 2024)), and channel-wise scaling (like in activation aware quantization (AWQ (Lin et al., 2024b)) or Smoothquant (Xiao et al., 2023)). Here, we propose a new transformation of the weight matrix using our

dual-scaling parameterization.

### 2.2.1. PSEUDO-ACTIVATION-AWARE QUANTIZATION FROM WEIGHT STRUCTURE

Prior work assumed that activation-awareness requires calibration data. Here, we find that analysis of the weight matrix can reveal information about the training data. Namely, we study the relationship between the per-column weight standard deviation $\vec{s}_{\mathbf{W}}$ and the sample mean of the absolute value of the input to a given layer, $\vec{\mu}_x$. We find that they have a strong predictive power of each other (see Fig. 2a and in appendix Fig. 6). We observe a similar relationship in all models we tested (incl. Qwen, Phi and Llama models). Notably, $\vec{\mu}_x$ is the quantity used in AWQ (Lin et al., 2024b) to determine pre-quantization column scales (see Sec. 2.2.2).

The underlying driver of this relation is that during training, weight updates are a function of the outer product of inputs and gradients. For simplicity, consider a single linear layer trained with input $\vec{x} \sim \mathcal{N}(\vec{0}, \vec{s}_x)$ using Adam (Kingma & Ba, 2014) on a noisy (Gaussian) target. In this setting, we observe (see Fig. 2b)

$$\vec{s}_{\mathbf{W}} \propto \frac{1}{\sqrt{|\vec{s}_x|}}. \tag{4}$$

However, scaling each column by its inverse standard deviation directly is a suboptimal strategy for quantization. This naive scaling often worsens per-*row*-outliers, which is a source of new quantization error (e.g. evidenced by an increased per-row kurtosis, see Fig. 2c). Thus, we must balance two effects: normalize column standard deviations to approximate activation-aware quantization, while simultaneously keeping row standard deviations approximately constant to avoid creating a new source of quantization errors. To achieve this, we propose a Sinkhorn-Knopp-style algorithm to iteratively normalize row and column standard deviations (Alg. 1, further implementation details in code in supplementary).

This Algorithm is essentially a search to minimize the matrix imbalance $I$, which we define as

$$I(\mathbf{W}) = \frac{\max\left(\max_i \sigma_i^{\text{row}}, \max_j \sigma_j^{\text{col}}\right)}{\min\left(\min_i \sigma_i^{\text{row}}, \min_j \sigma_j^{\text{col}}\right)} \tag{5}$$

where $\sigma_i^{\text{row}}$ and $\sigma_j^{\text{col}}$ denote the standard deviation of the $i$-th row and $j$-th column of $\mathbf{W}$, respectively.

Empirically, we find that this algorithm not only successfully balances row and column standard deviations, but that the input-side scales $\vec{t}$ it yields have a higher coefficient of determination with $\vec{\mu}_x$ than the pure standard deviation $\vec{s}_{\mathbf{W}}$.

Further evidence that SINQ exhibits a calibration-free activation-awareness, is that it does not necessarily reduce

---

**Algorithm 1** SINQ: A dampened Log-Space Sinkhorn-inspired Normalization. Iteratively normalize the standard deviation of the rows and columns of the matrix to be quantized. Then apply a standard quantization method.

---

**Require:** Weight matrix $\mathbf{W} \in \mathbb{R}^{m \times n}$, Iterations $K$, Bits $b$, Step-Sizes $s_{\min}, s_{\max}$
**Ensure:** Quantized weights $\mathbf{Q}$, Scales $\vec{s} \in \mathbb{R}^m, \vec{t} \in \mathbb{R}^n$
1: $\vec{\sigma}_{\text{row}} \leftarrow \vec{\sigma}^{\text{row}}(\mathbf{W}); \quad \vec{\sigma}_{\text{col}} \leftarrow \vec{\sigma}^{\text{col}}(\mathbf{W})$
2: $\tau \leftarrow \min(\min(\vec{\sigma}_{\text{row}}), \min(\vec{\sigma}_{\text{col}}))$ {Target variance}
3: $\mathbf{u} \leftarrow \vec{0}_m; \quad \mathbf{v} \leftarrow \vec{0}_n$ {Initialize log-scales}
4: $\theta^* \leftarrow \{\mathbf{u}, \mathbf{v}\}; \quad I_{\text{best}} \leftarrow \infty$ {Track best solution}
5: **for** $k \leftarrow 1$ to $K$ **do**
6: $\quad \hat{\mathbf{W}} \leftarrow (\mathbf{W} \oslash \exp(\mathbf{u})) \oslash \exp(\mathbf{v})$ {Apply current scales}
7: $\quad I_{\text{curr}} \leftarrow \text{Imbalance}(\hat{\mathbf{W}})$
8: $\quad$ **if** $I_{\text{curr}} < I_{\text{best}}$ **then**
9: $\quad\quad I_{\text{best}} \leftarrow I_{\text{curr}}; \quad \theta^* \leftarrow \{\mathbf{u}, \mathbf{v}\}$ {Snapshot best scales}
10: $\quad$ **end if**
11: $\quad \vec{\delta}_{\text{col}} \leftarrow \log\left(\text{clamp}(\vec{\sigma}^{\text{col}}(\hat{\mathbf{W}})/\tau, \ s_{\min}, \ s_{\max})\right)$
12: $\quad \vec{\delta}_{\text{row}} \leftarrow \log\left(\text{clamp}(\vec{\sigma}^{\text{row}}(\hat{\mathbf{W}})/\tau, \ s_{\min}, \ s_{\max})\right)$
13: $\quad \mathbf{v} \leftarrow \mathbf{v} + \vec{\delta}_{\text{col}}$ {Update col log-scales}
14: $\quad \mathbf{u} \leftarrow \mathbf{u} + \vec{\delta}_{\text{row}}$ {Update row log-scales}
15: **end for**
16: $\vec{s}, \vec{t} \leftarrow \exp(\mathbf{u}^*), \exp(\mathbf{v}^*)$ {Recover best linear scales}
17: $\hat{\mathbf{W}} \leftarrow (\mathbf{W} \oslash \vec{s}) \oslash \vec{t}$
18: $\mathbf{Q}, \vec{z}, \vec{s}_q \leftarrow \text{RoundToNearest}(\hat{\mathbf{W}}, b)$
19: **return** $\mathbf{Q}, \vec{z}, (\vec{s}_q \odot \vec{s}), \vec{t}$

---

the matrix reconstruction error, but it does reduce the activation reconstruction error of the output of the quantized layer. This stands in contrast to the standard uncalibrated space-transformation method: Hadamard rotation. In Fig. 3 we show precisely this: Hadamard rotation yields a better matrix reconstruction, but sometimes increases the output activation error over RTN. With SINQ, it is the other way around.

### 2.2.2. ACTIVATION-AWARE CALIBRATION WITH SINQ

Applying the proposed normalization of Alg. 1 is also helpful in combination with calibration. Consider, for example, AWQ (Lin et al., 2024b).

AWQ finds a vector of scales for each input of a linear layer by minimizing the 2-norm between the linear layer's output with the original and the scaled, quantized weight matrix. Formally,

$$\alpha^* = \text{argmin}_\alpha \left\| \vec{x} \cdot \mathbf{W}^{\mathsf{T}} - (\vec{x} \oslash \vec{\mu}_x^\alpha) \cdot d_q(q(\vec{\mu}_x^\alpha \odot \mathbf{W}))^{\mathsf{T}} \right\|_2, \tag{6}$$

where $\vec{x}$ is a set of inputs, $\vec{\mu}_x$ is the sample mean of the absolute value of $\vec{x}$, $q(\cdot)$ is the quantization function, $d_q(\cdot)$ is the dequantization function and $\alpha^*$ is a per-layer parameter (a scalar).[1] The final scale used is $\vec{\mu}_x^{\alpha^*}$ (i.e. the minimizer

---

[1] For results in combination with our method, we modify this formula by changing the norm to a 1-norm, which we observe to

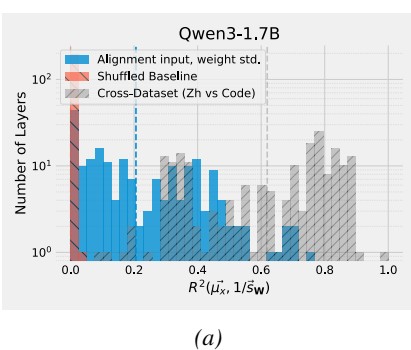

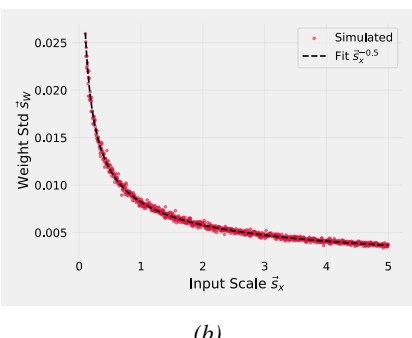

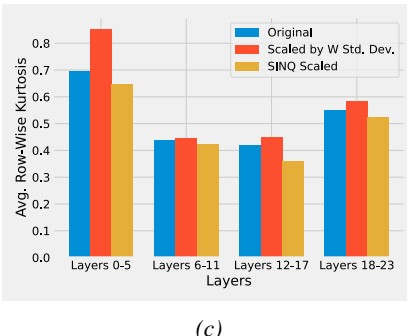

| (a) | (b) | (c) |

*Figure 2.* (a) In LLMs (here: Qwen3-1.7B) the reciprocal column-wise weight std. dev. is a good predictor of the average input magnitude (in terms of $R^2$-value), indicating a pseudo-activation-awareness. For comparison, the correlation when one vector is shuffled as a first baseline (that should be exceeded) and the cross-dataset correlation of average input magnitudes as a second baseline, Chinese wiki vs. Python code, as a plausible upper bound. (b) In a simplified setting (a single layer model), Adam training yields a simple relation between weight std. dev. (per column) and input scales. The underlying reason is that weight updates depend on the outer product of incoming inputs and gradients. (c) Directly scaling weights with their column-wise std. dev. creates more outliers row-wise, evidenced by increased kurtosis. SINQ successfully avoids this increase, while still scaling columns with a correlate of input scales.

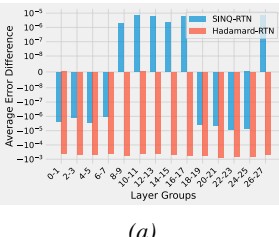

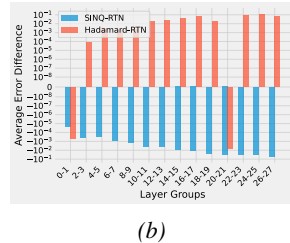

| (a) | (b) |

*Figure 3.* (a) Matrix reconstruction error with SINQ and Hadamard rotation plus RTN compared to RTN; y-axis: $E_i - E_{rtn}$, where $i$ is either SINQ or Hadamard (negative values: $i$ improves over RTN, positive values: $i$ is worse than RTN). (b) Activation reconstruction error in similar setting. In all cases Hadamard gives better matrix reconstruction, but SINQ better activation reconstruction in most cases. Evaluated on attention layers in Qwen3-1.7B.

of the activation reconstruction error).

We find that normalizing first with SINQ in this setting helps prevent the row-wise kurtosis from increasing too much when applying AWQ scales (similar to Fig. 2c), see Fig. 7. We term the combination of SINQ with AWQ 'ASINQ'.

### 2.3. Implementation Considerations

The second scale $\vec{t}$ can be applied as a scale vector to the input of the quantized linear layer, rather than when reconstructing the weight (see Eq. 7). In this formulation, the forward complexity of the dual-scaling approach becomes very similar to AWQ: The term inside the square bracket is the RTN dequantization, and for each linear layer, we need to do one additional element-wise scaling of activations (just

---

give slightly better results in combination with SINQ.

like in AWQ).

$$
\begin{aligned}
\vec{x} \cdot \mathbf{W}_{\text{approx}}^{\mathsf{T}} &= \vec{x} \cdot \left[ \vec{s} \odot (\mathbf{Q} + \vec{z}) \odot \vec{t} \right]^{\mathsf{T}} \\
&= \left( \vec{x} \odot \vec{t} \right) \cdot \left[ \vec{s} \odot (\mathbf{Q} + \vec{z}) \right]^{\mathsf{T}}
\end{aligned} \tag{7}
$$

The overhead of doing the additional scaling is small in practice, see Sec. 3.4 .

#### 2.3.1. NO-OVERHEAD SINQ

To avoid the small overhead of additional element-wise scaling, we can absorb input scales into preceding layers. This comes with the caveat that for many commonly used models, some layers need to share this second scale. For example, in Qwen-3 models, the Q, K, and V-layers share an input scale and the Gate and Up-Projection share an input scale. In the experiments, we show that this implies a trade-off between output quality (Appendix A.5) and a minor inference time overhead (see Sec. 3.4). No-overhead SINQ can also be used as a pure pre-processing step to improve the performance of other post-training quantization methods. E.g., for GGUF, we see improved accuracy while preserving end-to-end inference speed-up over fp16, see Appendix A.6.

## 3. Experiments

We evaluate our proposed methods against several strong baselines in 4-bit (and to a lesser extent 3-bit) quantization using the permissively licensed and powerful Qwen3 family of models by (Yang et al., 2025). We use the evaluation settings of (Zheng et al., 2025). In accordance with (Dutta et al., 2024), we report perplexities for language modeling and flip percentages for QA tasks. Flip percentages indicate how often the quantized model predicts a different result

*Table 1.* Weight-only uncalibrated uniform PTQ on Qwen3 models with 3-bit and 4-bit quantization, reporting perplexity and actual memory usage (GB). Lower is better for all metrics. The best result for a given setting is marked in **bold**.

| | Method | Qwen3-1.7B | | | Qwen3-14B | | | Qwen3-32B | | |
|---|---|---|---|---|---|---|---|---|---|---|
| | | Mem. | *Wiki2*↓ | *C4*↓ | Mem. | *Wiki2*↓ | *C4*↓ | Mem. | *Wiki2*↓ | *C4*↓ |
| | Original (BF16) | 3.44 | *16.67* | *19.21* | 29.54 | *8.64* | *12.01* | 65.52 | *7.60* | *10.77* |
| 3-BIT | RTN[†] | 1.28 | 32.43 | 31.10 | 9.23 | 10.50 | 14.88 | 17.61 | 30.78 | 35.83 |
| | Hadamard + RTN[†] | 1.28 | 32.40 | 31.07 | 9.23 | 10.60 | 15.10 | 17.61 | 11.26 | 14.83 |
| | HQQ | 1.28 | 32.10 | 30.54 | 9.23 | 10.73 | 14.39 | 17.62 | 9.09 | 12.58 |
| | **SINQ** *(ours)* | 1.28 | **22.39** | **24.88** | 9.25 | **9.33** | **12.90** | 17.61 | **8.79** | **11.83** |
| 4-BIT | RTN[†] | 1.42 | 18.74 | 20.81 | 10.54 | 8.95 | 12.50 | 20.78 | 8.92 | 12.80 |
| | Hadamard + RTN[†] | 1.42 | 19.10 | 20.70 | 10.54 | 8.85 | 12.35 | 20.78 | 8.28 | 11.60 |
| | HQQ | 1.42 | 18.96 | 22.10 | 10.54 | 8.78 | 12.36 | 20.78 | 8.62 | 12.20 |
| | **SINQ** *(ours)* | 1.42 | **17.14** | **19.83** | 10.56 | **8.76** | **12.21** | 20.73 | **7.74** | **10.96** |

[†] Baseline result obtained by running our own implementations.

from the original full-precision model. Additionally, benchmark results for reasoning benchmarks are provided in the appendix. Code to reproduce the perplexities reported for our methods in this section is available in the supplementary. We highlight that SINQ is architecture agnostic, i.e., there is no interdependency between the quantization of different layers (unlike, e.g., in methods using Hadamard transformations). For all models we tried, it works out of the box. Wherever there is no mention to the contrary, we set the group size to 64, batch-size to 8, and for SINQ use dual-scaling + shift parameterization.

To fairly account for the overhead of different parameterizations and tiling strategies, we report total memory use (including activations) in our experiments.

### 3.1. Uncalibrated Uniform Quantization

In Tab. 1, our method outperforms the baselines in every uncalibrated case in terms of C4 (Raffel et al., 2020) and WikiText2 perplexity, sometimes reducing the residual difference to the 16-bit baseline by more than half. Similarly, our method performs best in terms of the average number of flips (see Tab. 2). Fig. 4 shows the memory-perplexity Pareto plot for different quantization methods across a wide range of Qwen3 models. Because the Qwen3 models are available in many different sizes, our method can dominate the bfloat16 baselines across a large range of available memory, from ca. 1.5 GB to 65 GB. Additional perplexity results on Llama (Sec. A.8), DeepSeek-V3 (Sec. A.9), Phi (Sec. A.11) and other Mixture-of-Experts (MoE, (Fedus et al., 2022)) models (Sec. A.15) can be found in the Appendix.

#### 3.1.1. RESULTS ON LARGE MODELS

We further evaluate our method on two large models, Qwen3-235B-A22B by (Yang et al., 2025) and DeepSeek-

V2.5-236B (DeepSeek-AI, 2024), see Tab. 13 in the appendix. Notably, these are both MoE models, and the latter uses Multi-head Latent Attention (MLA). This underlines the robustness of SINQ to different architectures.

### 3.2. Uncalibrated Non-Uniform Quantization

SINQ is compatible with non-uniform quantization levels, for example, NF4 as defined by (Dettmers et al., 2023). In Tab. 3 we compare to various non-uniform 4-bit quantization methods. We simply replace the RoundToNearest function in Alg.1 with the NF4 quantizer. Also, here the SINQ method improves over the NF4 baseline. We note that for the 32B model, SINQ with INT4 slightly outperforms SINQ with NF4.

### 3.3. Calibrated Uniform Quantization

To demonstrate compatibility with calibration approaches, in Tab. 4 we consider the combination of SINQ and AWQ (see Sec. 2.2.2 for the methodology). For a better match to the original AWQ implementation, we quantize our $\vec{s}, \vec{z}$ to 8 bits in these calibrated experiments. In several cases, even our uncalibrated method outperforms the calibrated baselines, but the addition of AWQ calibration brings further improvements.

While our weight structure-based estimation of input scales is effective, calibration data is still useful and provides additional information.

### 3.4. Inference Time

The inference time of SINQ-quantized models is very close to, or identical to, that of models quantized with standard methods like HQQ or GPTQ. Specifically, the no-overhead formulation of SINQ (see Sec. 2.3.1) achieves identical inference time and we report end-to-end results on `llama.cpp`

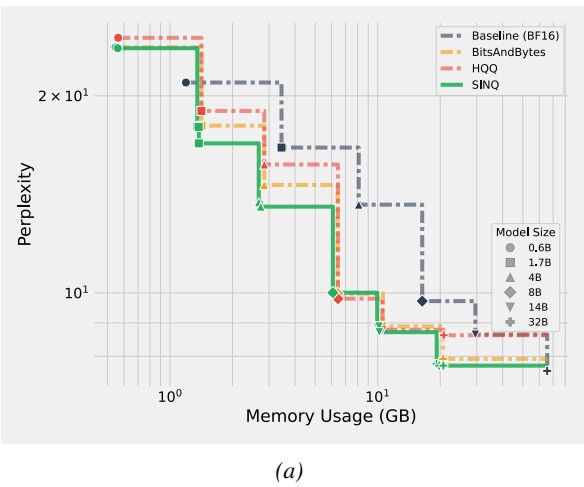
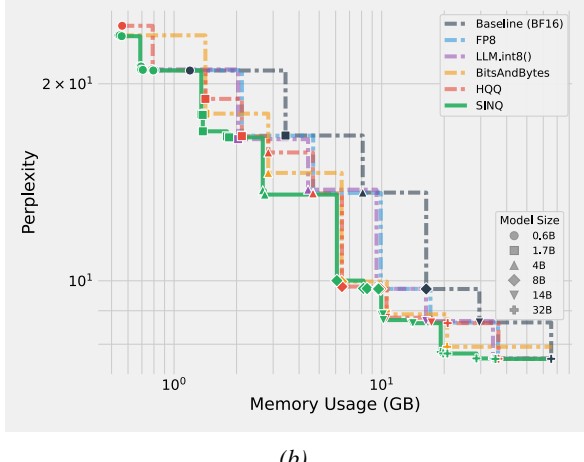

*(a)*      *(b)*

*Figure 4.* Pareto plot in terms of memory vs. WikiText2 perplexity for Qwen3-0.6B to 32B for different uncalibrated quantization methods. (a) compares different 4-bit methods (including FP4, INT4, and NF4 where available). The maximum distance from the 4-bit pareto front of our method is $< 0.01$ppl. Note that the difference to the baseline is small. (b) allows bit widths of 4, 6, 8. For 8-bit quantization we inlcude `LLM.int8()` from (Dettmers et al., 2022) as a reference method. Both plots include the BF16 model as a baseline. For these plots we allow group sizes 64 and 128 for all methods.

in Appendix A.6.

For the standard SINQ formulation (with dual-scale overhead), we compare the inference time of a HQQ-quantized linear layer using the **gemlite** kernel (Badri et al., 2024) with that of a SINQ-quantized layer. For the latter, we implement the second scale using a PyTorch element-wise multiply before applying the kernel. As shown in Tab. 5, this simple approach incurs a negligible $1.8\%$ overhead in the most challenging setting of batch size 1. We additionally report end-to-end inference results in `SGLang` (Zheng et al., 2024) in Tab. 6.

### 3.5. Quantization Time

Quantization with SINQ is fast. On identical hardware, SINQ has an average runtime of $1.1\times$ our RTN baseline. This is faster than the already efficient HQQ, at $> 2\times$, or calibrated methods like AWQ, at $> 30\times$ the RTN baseline. Further details are given in Tab. 10 and Fig. 8 in the appendix.

### 3.6. Ablation Studies

We compare several variants of our method, 1) with and without shifts, 2) quantized (int8) and half precision (fp16) auxiliary variables. In Fig. 5, we see that in general, both precisions work well and both settings have their sections of the Pareto front. The use of shifts does improve the Pareto front appreciably. Based on these results, we use shifts and quantize the auxiliaries to match the methods we are comparing against.

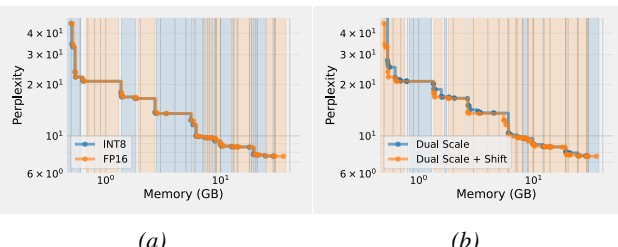

*(a)*      *(b)*

*Figure 5.* Ablation experiments in the form of memory-perplexity Pareto-fronts across the Qwen3 family. (a) Auxiliary variable precision (b) Using or not using shifts.

## 4. Related Work

### 4.1. Uncalibrated, Uniform Integer Quantization

Most closely related to our approach are works focusing on quantization to uniform integer values without the use of a calibration set. Beyond the trivial (but effective) round-to-nearest (RTN) method with scales and shifts chosen to cover the full range of the input weights, there have been two major innovations in this domain. Firstly, half-quadratic quantization (HQQ, (Badri & Shaji, 2023)) proposes optimizing the values of the shifts found by RTN, so that a $p$-norm (usually $p = 0.7$) error between the original and the quantized matrix becomes minimal. Secondly, applying a Hadamard transform to all weights in a network has been observed to normalize the weight distributions (Tseng et al., 2024a), which often eases quantization. The Hadamard approach has a high-level similarity to our approach, in that we also transform the weight matrices to find an easier-to-quantize format.

*Table 2.* Flip rates (%) (as proposed by (Dutta et al., 2024) since more reliable than accuracy) on HellaSwag, PIQA, and MMLU for Qwen3 models with 3-bit and 4-bit quantization. Lower is better. The best result for a given setting is marked in **bold**. Accuracy measurements are given in the appendix.

| | | | Qwen3-14B | | | | Qwen3-32B | | |
|---|---|---|---|---|---|---|---|---|---|
| | | Method | HellaSwag | PIQA | MMLU | Avg.↓ | HellaSwag | PIQA | MMLU | Avg.↓ |
| CALIBRATION-FREE | 3-BIT | RTN[†] | 8.44 | 8.60 | 10.97 | 9.34 | 22.84 | 17.08 | 10.61 | 16.84 |
| | | Hadamard + RTN[†] | 10.68 | 10.93 | 16.21 | 12.60 | 19.83 | 13.17 | 12.81 | 15.27 |
| | | HQQ | 7.99 | 7.94 | 14.28 | 10.07 | 7.23 | 9.30 | 10.98 | 9.17 |
| | | **SINQ** *(ours)* | **5.34** | **7.02** | **10.82** | **7.73** | **5.54** | **7.13** | **10.21** | **7.63** |
| | 4-BIT | RTN[†] | 2.92 | 4.57 | 4.89 | 4.13 | 4.18 | 6.31 | 5.28 | 5.26 |
| | | BnB (FP4) | 4.21 | 5.71 | 6.72 | 5.55 | 12.32 | 9.14 | 6.25 | 9.24 |
| | | BnB (NF4) | 2.66 | **3.10** | 4.70 | 3.49 | 3.73 | 3.48 | 4.76 | 3.99 |
| | | Hadamard + RTN[†] | 3.63 | 5.55 | 4.88 | 4.69 | 4.01 | 6.02 | 5.32 | 5.12 |
| | | HQQ | 2.81 | 4.35 | 5.17 | 4.11 | 5.83 | 5.18 | 4.98 | 5.33 |
| | | **SINQ** *(ours)* | **2.36** | **3.37** | **4.65** | **3.46** | **2.52** | **3.59** | **4.69** | **3.60** |
| CALIBRATED | 3-BIT | GPTQ | 5.18 | 7.83 | 11.17 | 8.06 | 6.33 | 8.76 | 10.25 | 8.45 |
| | | Hadamard[†] + GPTQ | 5.14 | 7.56 | 11.15 | 7.95 | 5.52 | 8.71 | **10.08** | 8.10 |
| | | **A-SINQ** *(ours)* | **5.13** | **7.18** | **10.36** | **7.56** | **5.23** | **7.62** | 10.15 | **7.67** |
| | 4-BIT | GPTQ | 2.24 | 4.13 | 4.56 | 3.64 | 2.78 | **3.48** | 4.80 | 3.69 |
| | | Hadamard[†] + GPTQ | 2.22 | 3.54 | 4.53 | 3.43 | 2.70 | 3.54 | 4.79 | 3.68 |
| | | AWQ | 2.23 | 3.26 | **4.10** | 3.20 | 2.59 | 4.13 | 4.44 | 3.72 |
| | | **A-SINQ** *(ours)* | **2.20** | **3.11** | 4.23 | **3.18** | **2.57** | 3.86 | **4.38** | **3.60** |

[†] Baseline result obtained by running our own implementations.

*Table 3.* Weight-only uncalibrated PTQ on Qwen3 models with 4-bit non-uniform quantization, reporting perplexity and actual memory usage (GB). Lower is better for all metrics. The best *non-uniform* result for a given setting is marked in **bold**, the results where SINQ with uniform quantization outperforms the non-uniform baselines are marked red.

| | | Qwen3-1.7B | | | Qwen3-14B | | | Qwen3-32B | | |
|---|---|---|---|---|---|---|---|---|---|---|
| | Method | Mem. | Wiki2↓ | C4↓ | Mem. | Wiki2↓ | C4↓ | Mem. | Wiki2↓ | C4↓ |
| | Original (BF16) | 3.44 | 16.67 | 19.21 | 29.54 | 8.64 | 12.01 | 65.52 | 7.60 | 10.77 |
| 4-BIT | BnB (FP4) | 1.42 | 24.05 | 23.44 | 10.59 | 8.88 | 12.54 | 20.67 | 11.93 | 16.90 |
| | BnB (NF4) | 1.42 | 18.00 | 20.43 | 10.59 | 8.89 | 12.27 | 20.67 | 7.94 | 11.21 |
| | HIGGS (non-uniform) | 1.51 | 23.98 | 25.27 | 10.28 | 9.13 | 12.56 | 19.88 | 8.02 | 11.24 |
| | **SINQ** (NF4) *(ours)* | 1.42 | **16.94** | **19.83** | 10.56 | **8.72** | **12.13** | 20.73 | **7.83** | **10.97** |
| | **SINQ** *(ours, uniform)* | 1.42 | 17.14 | 19.83 | 10.56 | 8.76 | 12.21 | 20.73 | 7.74 | 10.96 |

## 4.2. Non-Uniform Quantization

After training, neural network weights are usually not uniformly distributed. Therefore, quantization incurs lower errors when the quantization levels are also non-uniform, to match the distribution of the trained weights. (Dettmers et al., 2023) proposes quantiles of the normal distribution as a preferable set of quantization levels resulting in the normal-float-4 (NF4) format (in the 4-bit case). The variance between optimal levels across different layers in a network is reduced when the weights of the network have been Hadamard transformed. This is used in HIGGS by (Malinovskii et al., 2025) together with non-uniform quantization: Non-uniform quantization levels can be synergistic with weight matrix transformations. SINQ is orthogonal to the uniformity of the quantization levels; we show that it

is compatible with non-uniform quantization in NF4-based experiments.

## 4.3. Calibration

If quantization time and potential overfitting can be tolerated, using some data to calibrate the quantized value assignments can be a practical approach. A highly influential work is GPTQ (Frantar et al., 2022) that considers the Hessian for a given layer to find weight pairs that can compensate for each other, if their quantization errors have opposite signs. A second approach, as seen in AWQ (Lin et al., 2024b), is to minimize the prediction error of each linear layer (separately) under quantization (for more details see Sec. 2.2.2). This per-layer prediction error minimization has been further developed by (Shao et al.) and (Ma et al.).

*Table 4.* Weight-only PTQ on Qwen3 models with 3-bit and 4-bit quantization, reporting perplexity and actual memory usage (GB). Lower is better for all metrics. The best result for a given setting is marked in **bold**, the *calibration-free* results that outperform all calibrated baselines at equal bits (other than our own) are marked red.

| | Method | Qwen3-1.7B | | | Qwen3-14B | | | Qwen3-32B | | |
|---|---|---|---|---|---|---|---|---|---|---|
| | | Mem. | *Wiki2*↓ | *C4*↓ | Mem. | *Wiki2*↓ | *C4*↓ | Mem. | *Wiki2*↓ | *C4*↓ |
| | Original (BF16) | 3.44 | *16.67* | *19.21* | 29.54 | *8.64* | *12.01* | 65.52 | *7.60* | *10.77* |
| 3-BIT | GPTQ | 1.26 | 32.21 | 31.05 | 9.28 | 9.54 | 13.03 | 17.70 | 9.03 | 12.38 |
| | Hadamard[†] + GPTQ | 1.26 | 24.70 | 25.37 | 9.28 | 9.61 | 12.92 | 17.70 | 8.51 | 11.63 |
| | **A-SINQ** *(ours)* | 1.26 | **22.30** | **24.00** | 8.90 | **9.31** | **12.71** | 16.68 | **8.45** | **11.54** |
| | **SINQ** *(ours, calibration-free)* | 1.28 | 22.39 | 24.88 | 9.25 | 9.33 | 12.90 | 17.61 | 8.79 | 11.83 |
| 4-BIT | GPTQ | 1.38 | 19.70 | 21.51 | 10.24 | 8.81 | 12.22 | 19.99 | 7.80 | 10.99 |
| | Hadamard[†] + GPTQ | 1.38 | 18.12 | 20.38 | 10.24 | 8.81 | 12.19 | 19.99 | 7.78 | 10.95 |
| | AWQ | 1.38 | 16.90 | 19.95 | 10.25 | 8.78 | 12.24 | 20.00 | 7.79 | 10.96 |
| | **A-SINQ** *(ours)* | 1.38 | **16.67** | **19.73** | 10.21 | **8.71** | **12.13** | 19.83 | 7.78 | **10.93** |
| | **SINQ** *(ours, calibration-free)* | 1.42 | 17.14 | 19.83 | 10.58 | 8.76 | 12.21 | 20.73 | 7.74 | 10.96 |

[†] Baseline result obtained by running our own implementations.

*Table 5.* Computational overhead of the additional scale in a naive implementation. We compare the matmul speed of the fast gemlite kernel for W4A16 operation with and without the additional scale as used by SINQ. In practice, this scale can often be absorbed into other operations to reduce overhead further. **B** indicates batchsize, **D** the input/output dimension, and **g**(·) is the gemlite kernel.

| **B** | **D** | **g**($\vec{x}$) [ms] | **g**($\vec{x} \cdot \vec{t}$) [ms] | **Overhead [%]** |
|---|---|---|---|---|
| 1 | 1024 | 0.0446 | 0.0454 | 1.8% |
| 1 | 2048 | 0.0448 | 0.0455 | 1.5% |
| 64 | 1024 | 0.0472 | 0.0476 | 0.8% |
| 64 | 2048 | 0.0479 | 0.0483 | 0.9% |

*Table 6.* Baseline W16A16 end-to-end decode throughput on SGLang (Zheng et al., 2024) in terms of tokens/s (tps, higher is better) at batch size 1 context length 256, generation length 512 as well as speedups over W16A16 for AWQ and SINQ at W4A16.

| Method | **Llama2-7B** | **Llama3.1-8B** | **Qwen3-14B** | **Qwen3-32B** |
|---|---|---|---|---|
| FP16 | *96tps* | *86tps* | *48tps* | *21tps* |
| AWQ | 2.4× | 2.2× | 2.4× | 2.9× |
| SINQ | 2.3× | 2.1× | 2.4× | 2.8× |

Similar to AWQ, CrossQuant (Liu et al., 2024b) finds an input axis scale for the weight matrix with a calibration process. (Elhoushi & Johnson, 2025) combine non-uniform quantization with calibration to learn optimal non-uniform quantization levels. We demonstrate the compatibility of SINQ with calibration in AWQ-based experiments.

### 4.4. Weight Space Transformations

The concept of weight space transformation, such as applying the Hadamard transform, a random rotation, or scaling with a diagonal matrix, can be further improved by combining it with calibration and/or non-uniform quantization.

HIGGS (Malinovskii et al., 2025) applies Hadamard transforms and matches non-uniform quantization levels to the typically resulting distribution. QuaRot (Ashkboos et al., 2024), SpinQuant (Liu et al.), and FlatQuant (Sun et al.) combine various calibration methods with rotations (including the Hadamard transform). Duquant (Lin et al., 2024a) combines learned rotations with permutations for further flexibility. In Kurtail, (Akhondzadeh et al.) optimize rotations on a kurtosis proxy target. Several of these methods specifically target joint activation and weight quantization. The key differences to our method are that we use the dual-scaling and minimize the matrix imbalance, allowing the method to be uniform, calibration-free and, compared to rotated models, architecture agnostic (similar to HQQ (Badri & Shaji, 2023) and BnB (Dettmers et al., 2023)) in the sense that each linear layer can be treated independently (which is helpful for generalization to new architectures).

## 5. Conclusion

We have shown that the column-wise standard deviations of LLM weight matrices are predictive of input activations. To enable the use of this pseudo-activation-awareness for uncalibrated weight quantization, we have proposed a modified Sinkhorn iteration procedure that normalizes both row and column standard deviations, which successfully balances a pseudo-activation-aware column scaling, against row-wise kurtosis impact. We show in numerous experiments that this method is fast and outperforms state-of-the-art methods for uniform quantization without calibration, and can be combined with widely used calibrated and/or non-uniform methods. Finally, we demonstrate that the computational overhead of an additional scale is negligible at the individual-layer level and in end-to-end measurements.

## Impact Statement

This paper presents work whose goal is to advance the field of machine learning. There are many potential societal consequences of our work, none of which we feel must be specifically highlighted here.

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

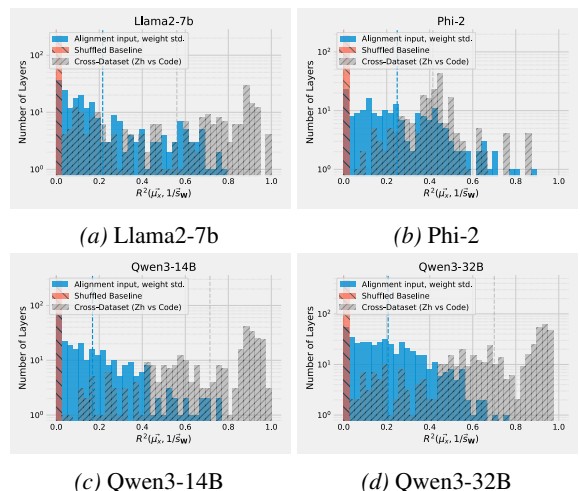

*(a)* Llama2-7b     *(b)* Phi-2

*(c)* Qwen3-14B     *(d)* Qwen3-32B

*Figure 6.* $R^2$-Values between activations and weight standard deviations (column-wise) in different models.

Zheng, X., Li, Y., Chu, H., Feng, Y., Ma, X., Luo, J., Guo, J., Qin, H., Magno, M., and Liu, X. An empirical study of qwen3 quantization. *arXiv preprint arXiv:2505.02214*, 2025.

# A. Appendix

## A.1. Reproducibility

The code used to derive our LLM quantization results is given in the supplementary. This includes a full implementation of our method. For our key results, the perplexity evaluations, we use open-source code by (Zheng et al., 2025) to ensure reproducible detail settings (e.g., context length). Our code, as well as the external code we base ours on, is permissively licensed to facilitate follow-up research. For our experiments, we use permissively licensed open-weight models to promote reproducibility further.

## A.2. Weight Standard Deviation Plots for Further Models

In Fig. 6 we see that also in LLama2, Phi-2 and further Qwen3 models, the column-wise weight standard deviation and the input magnitude show a high $R^2$-value.

## A.3. AWQ vs. ASINQ Row-Wise Kurtosis

In Fig. 7 we show the average row-wise kurtosis for AWQ and ASINQ on Qwen3-1.7B. We see that using SINQ before AWQ reduces the row-wise kurtosis, which explains the better quantized performance. The average kurtosis reduction is $1.32\times$.

*Table 7.* Reasoning performance on Qwen3-14B with 4-bit weight-only PTQ.

| | | | **Qwen3-14B** | | | | | |
|---|---|---|---|---|---|---|---|---|
| | | Method | *AIME 2024* | | *AIME 2025* | | Avg. | |
| | | | Tok. | Acc. (%)↑ | Tok. | Acc. (%)↑ | Δ Tok. | Acc. (%)↑ |
| | | Original (FP16) | *11 464* | *76.70* | *12 636* | *63.30* | *0* | *70.00* |
| CALIBRATION-FREE | 4-BIT | RTN | 10 973 | 66.70 | 12 642 | 50.00 | -242 | 58.35 |
| | | BnB (FP4) | 11 500 | 60.00 | 12 455 | 53.30 | -72 | 56.65 |
| | | BnB (NF4) | 12 132 | 70.00 | 12 899 | 56.70 | +930 | 63.35 |
| | | Hadamard + RTN | 11 210 | 70.00 | 12 989 | 53.30 | +99 | 61.65 |
| | | HQQ | 11 862 | 70.00 | 12 991 | 56.70 | +367 | 63.35 |
| | | **SINQ** | 11 660 | **73.30** | 12 305 | **63.30** | -67 | **68.30** |

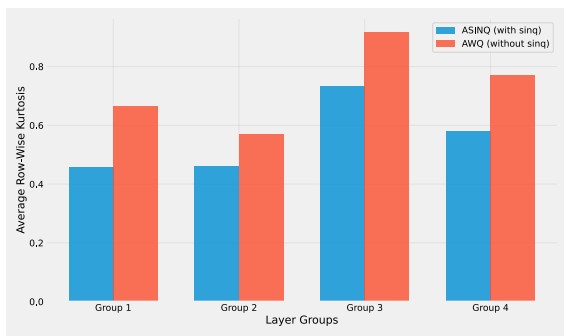

*Figure 7.* Average row-wise kurtosis for AWQ and ASINQ across different layer groups in Qwen3-1.7B. Using SINQ prevents kurtosis increase in AWQ scaling.

## A.4. Results on Reasoning

In Tab. Table 7 we show results on reasoning benchmarks (Ye et al., 2025). Here, we include the length of reasoning traces to ensure that lengthened reasoning does not negate some of the upside of quantization. Note that these are preliminary pass@1 results. These preliminary findings seem to suggest that the proposed method sustains robust reasoning capabilities while avoiding an increase in reasoning trace length, which is crucial for preserving the efficiency gains achieved through quantization.

## A.5. No-Overhead variant

In Tab. 8, we show that the overhead-free formulation of SINQ also produces better quality outputs than comparable prior methods.

## A.6. Improving GGUF with no-overhead SINQ

Table 9 demonstrates that no-overhead SINQ reduces perplexity over standard GGUF while preserving inference speedups, evaluated on `Q4_0` and `Q3_KS`, the prevalent GGUF formats for 4-bit and 3-bit quantization.

## A.7. Timing Results

In Tab. 10 we report quantization time results on a single GPU for various models. Although precise timings may vary with hardware, our method achieves times comparable to the RTN baseline and even surpasses HQQ, which is already regarded as a fast quantization technique. Furthermore, the calibrated version, A-SINQ, is substantially faster than popular state-of-the-art calibrated methods like GPTQ and AWQ. Fig. 8 shows the distribution of quantization times over 10 runs for various popular quantization methods on Qwen3-32B on GPU.

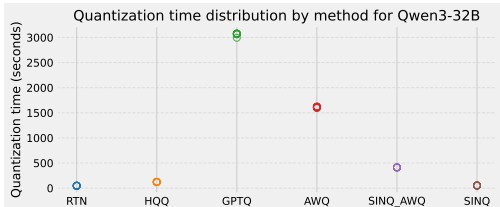

*Figure 8.* Distribution of quantization times for each method for Qwen3-32B.

## A.8. Results on Llama Models

In Tab. 11 we report quantization results on Llama family models. These findings further validate the effectiveness of SINQ also on this type of architecture.

## A.9. Results on DeepSeek-V3 and other large models

In Tab. 12 we compare HQQ to SINQ on WikiText2 perplexity for DeepSeek-V3 (Liu et al., 2024a) while in Tab. 13 we report the results for DeepSeek-V2.5-236B and Qwen3-235B-A22B.

*Table 8.* No-Overhead SINQ variant. Weight-only uncalibrated uniform PTQ on Qwen3 models with 4-bit quantization, reporting perplexity and actual memory usage (GB). Lower is better for all metrics. The best result for a given setting is marked in **bold**.

| Method | Qwen3-1.7B | | | Qwen3-14B | | | Qwen3-32B | | |
|---|---|---|---|---|---|---|---|---|---|
| | Mem. | *Wiki2↓* | *C4↓* | Mem. | *Wiki2↓* | *C4↓* | Mem. | *Wiki2↓* | *C4↓* |
| Original (BF16) | 3.44 | *16.67* | *19.21* | 29.54 | *8.64* | *12.01* | 65.52 | *7.60* | *10.77* |
| Hadamard + RTN[†] | 1.42 | 19.10 | 20.70 | 10.54 | 8.85 | 12.35 | 20.78 | 8.28 | 11.60 |
| HQQ | 1.42 | 18.96 | 22.10 | 10.54 | 8.78 | 12.36 | 20.78 | 8.62 | 12.20 |
| **SINQ** *(ours)* | 1.42 | **17.14** | **19.83** | 10.56 | **8.76** | **12.21** | 20.73 | **7.74** | **10.96** |
| **SINQ no overhead** *(ours)* | 1.42 | 17.63 | 19.99 | 10.56 | 8.78 | 12.32 | 20.73 | 7.78 | 11.15 |

[†] Baseline result obtained by running our own implementations.

*Table 9.* Performance of 4-bit and 3-bit no-overhead SINQ combined with GGUF quantization in `llama.cpp` on GPU, evaluated at batch size 512 (input length 512, output length 128). We report perplexity (Ppl., lower is better) measured on WikiText-2, as well as prefill and decode throughput (tokens/s, higher is better) measured on a 512-token prompt. Speedups are reported relative to the FP16 baseline. No-overhead SINQ+GGUF consistently improves GGUF quantization.

| Method | Qwen3 0.6B | | | Qwen3 1.7B | | | Qwen3 4B | | |
|---|---|---|---|---|---|---|---|---|---|
| | Ppl.↓ | Prefill ↑ | Decode ↑ | Ppl.↓ | Prefill ↑ | Decode ↑ | Ppl.↓ | Prefill ↑ | Decode ↑ |
| *Base (FP16)* | *21.88* | *22034 tps* | *647 tps* | *17.13* | *12762 tps* | *346 tps* | *14.31* | *5889 tps* | *167 tps* |
| Base + Q4_0 | 25.25 | 1.93× | 1.31× | 21.30 | 2.56× | 1.80× | 14.97 | 2.92× | 2.10× |
| No-over. SINQ + Q4_0 | **24.01** | 1.93× | 1.31× | **17.65** | 2.56× | 1.80× | **14.77** | 2.92× | 2.10× |
| Base + Q3_KS | 35.59 | 1.77× | 1.16× | 24.02 | 2.19× | 1.62× | 19.03 | 2.55× | 1.80× |
| No-over. SINQ + Q3_KS | **30.20** | 1.78× | 1.16× | **19.49** | 2.19× | 1.62× | **18.06** | 2.54× | 1.80× |

*Table 12.* Weight-only PTQ on **DeepSeek-V3-685B** with 4-bit quantization. We report perplexity on WikiText-2 (lower is better). Best per setting in **bold**.

| Setting | Method | *Wiki2↓* |
|---|---|---|
| Calibration-free (4-bit) | HQQ | 5.38 |
| | SINQ | **5.31** |

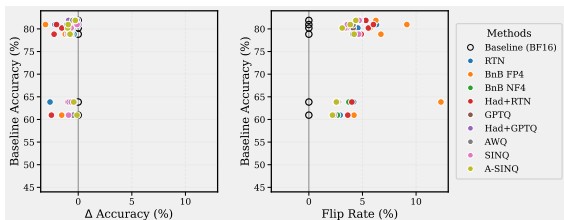

*Figure 9.* Comparison of baseline accuracy, accuracy changes, and flip rates across different 4-bit quantization methods (similar to (Dutta et al., 2024)). On QA tasks, flips have been shown to be the more consistent quality metric of LLM quantization.

## A.10. Accuracy Results

In Fig. 9 and Tab. 14 we report accuracy results on various QA tasks. Note that flips (as reported in the main paper) are the more reliable (and less easily manipulated) metric than accuracy for QA tasks, as shown in (Dutta et al., 2024). Fig. 9 closely follows the analysis presented in prior work (Dutta et al., 2024), further confirming the alignment of our findings with existing literature.

## A.11. Results on Phi Models

In Tab. 15 we report quantization results on Phi family models. These findings further validate the effectiveness of SINQ also on this type of architecture.

## A.12. Comparison to CrossQuant in W4A8 setting

Here we compare to the CrossQuant method (Liu et al., 2024b). We separate these results from the main text, because CrossQuant uses a W4A8G128 setting, so that the values are not directly comparable to the main results (using W4A16G64) of the paper. See Tab. 16.

*Table 10.* Average quantization time (seconds) across 10 runs for some Qwen3 models on GPU, comparing different quantization methods. The rightmost column reports the relative average slowdown with respect to RTN.

| Method | Qwen3-1.7B | Qwen3-4B | Qwen3-8B | Qwen3-14B | Qwen3-32B | Avg. cost |
|---|---|---|---|---|---|---|
| *RTN* | *2.91 s ±0.11* | *6.32 s ±0.06* | *11.35 s ±0.31* | *20.61 s ±0.87* | *46.79 s ±2.52* | *1.00×* |
| HQQ | 3.65 s ±0.13 | 10.15 s ±0.27 | 24.06 s ±1.54 | 43.62 s ±0.54 | 122.45 s ±2.45 | 2.32× |
| GPTQ | 193.33 s ±1.68 | 426.89 s ±0.75 | 669.06 s ±0.84 | 1160.37 s ±1.68 | 3064.62 s ±24.33 | 62.68× |
| AWQ | 104.63 s ±9.26 | 225.27 s ±3.91 | 392.51 s ±2.86 | 695.29 s ±1.19 | 1613.75 s ±9.79 | 34.46× |
| A-SINQ *(ours)* | 23.86 s ±0.17 | 49.81 s ±0.13 | 92.17 s ±0.33 | 173.93 s ±0.38 | 411.95 s ±0.57 | 8.54× |
| **SINQ** *(ours)* | **3.03 s ±0.29** | **6.33 s ±0.52** | **13.23 s ±0.64** | **21.38 s ±2.15** | **51.56 s ±2.00** | **1.09×** |

*Table 11.* Weight-only PTQ on Llama models with 3-bit and 4-bit quantization, reporting perplexity and actual memory usage (GB). Lower is better for all metrics. In bold is the best result for a given setting.

| | | | Llama 2-7B | | | Llama 3-8B | | | Llama 3-70B | |
|---|---|---|---|---|---|---|---|---|---|---|---|
| | | Method | Mem. | *Wiki2↓* | *C4↓* | Mem. | *Wiki2↓* | *C4↓* | Mem. | *Wiki2↓* | *C4↓* |
| | | Original (BF16) | 14.08 | *5.47* | *6.90* | 17.45 | *6.13* | *9.61* | 141.11 | *2.86* | *7.30* |
| CALIBRATION-FREE | 3-BIT | RTN | 3.54 | 6.40 | 8.05 | 5.25 | 10.18 | 15.27 | 35.93 | 5.26 | 10.80 |
| | | Hadamard + RTN | 3.54 | 6.31 | 7.89 | 5.25 | 9.97 | 15.25 | 35.93 | 4.99 | 10.45 |
| | | HQQ | 3.62 | 7.05 | 9.03 | 5.24 | 9.55 | 14.68 | 36.16 | 85.64 | 23.32 |
| | | **SINQ** *(ours)* | 3.54 | **6.14** | **7.72** | 5.35 | **8.04** | **12.32** | 35.93 | **4.52** | **8.48** |
| | 4-BIT | RTN | 4.17 | 5.67 | 7.14 | 6.06 | 6.61 | 10.25 | 42.71 | 3.56 | 10.58 |
| | | BnB (FP4) | 4.17 | 5.76 | 7.24 | 6.06 | 6.93 | 10.75 | 42.71 | 3.58 | 8.23 |
| | | Hadamard + RTN | 4.17 | 5.65 | 7.10 | 6.06 | 6.72 | 10.23 | 42.71 | 3.54 | 9.95 |
| | | HQQ | 4.22 | 5.68 | 7.13 | 6.06 | 6.58 | 10.22 | 42.71 | 3.26 | 8.13 |
| | | **SINQ** *(ours)* | 4.19 | **5.60** | **7.04** | 6.06 | **6.53** | **10.14** | 42.81 | **3.17** | **7.51** |
| | | BnB (NF4) | 4.17 | 5.65 | 7.09 | 6.07 | 6.56 | 10.20 | 42.71 | 3.22 | 7.68 |
| | | **SINQ** (NF4) *(ours)* | 4.18 | **5.58** | **7.03** | 6.07 | **6.51** | **10.09** | 42.81 | **3.16** | **7.50** |

*Table 13.* Weight-only PTQ on **DeepSeek-V2.5-236B** and **Qwen3-235B-A22B** MoE models with 3-bit and 4-bit quantization, reporting perplexity and actual memory usage (GB). Lower is better for all metrics. The best result for a given setting is marked in **bold**.

| Setting | Method | DeepSeek-V2.5-236B | | | Qwen3-235B-A22B | | |
|---|---|---|---|---|---|---|---|
| | | Mem. | *Wiki2*↓ | *C4*↓ | Mem. | *Wiki2*↓ | *C4*↓ |
| Baseline | Original (BF16) | 471.56 | 5.36 | 8.15 | 470.19 | 5.37 | 9.30 |
| Calibration-free (3-bit) | RTN | 110.90 | 5.91 | 8.84 | 110.98 | 10.11 | 13.92 |
| | HQQ | 110.92 | 5.89 | 8.76 | 114.43 | 13.07 | 16.38 |
| | **SINQ** *(ours)* | 110.91 | **5.82** | **8.74** | 110.99 | **6.27** | **10.03** |
| Calibration-free (4-bit) | RTN | 134.24 | 5.49 | 8.27 | 134.03 | 5.65 | 9.49 |
| | BnB (FP4) | 134.52 | 5.55 | 8.41 | 134.10 | 6.67 | 10.21 |
| | BnB (NF4) | 134.52 | 5.49 | 8.28 | 134.10 | 5.60 | 9.49 |
| | HQQ | 134.25 | 5.49 | 8.27 | 134.03 | 5.60 | 9.46 |
| | **SINQ** *(ours)* | 134.51 | **5.48** | **8.25** | 134.06 | **5.58** | **9.43** |

*Table 16.* Wikitext perplexity comparison to CrossQuant on Llama2 models (we use context length 2048, W4A8G128 to match reported CrossQuant results). Lower is better. In bold is the best result.

| | Llama2-7B | Llama2-13B |
|---|---|---|
| Method | *Wiki2*↓ | *Wiki2*↓ |
| Original (BF-16) | 5.47 | 4.88 |
| CrossQuant | 5.79 | 5.14 |
| ASINQ | **5.62** | **4.97** |

### A.13. Comparison to Code-Book-Based Methods

Here we compare to two recent code-book-based methods by (Tseng et al., 2024b) and (Tseng et al., 2024a). Note that code-book-based methods are incompatible with activation quantization and require non-standard operations / kernels (may not work on NPU, TPU, mobile). See Tab. 17.

*Table 17.* Wikitext perplexity comparison to code-book-based models on Llama2 models (context length 4096). Note that code-book-based methods are incompatible with activation quantization and require non-standard operations (may not be NPU, TPU, mobile compatible).

| | Llama2-7B | Llama2-13B |
|---|---|---|
| Method | *Wiki2*↓ | *Wiki2*↓ |
| Baseline | 5.12 | 4.57 |
| QTIP | **5.17** | **4.62** |
| QUIP# | 5.22 | 4.65 |
| ASINQ | 5.22 | 4.64 |

In Tab. 18 we compare end-to-end speed to QTIP. We see that code-book-based methods have additional overhead for dequantization as expected.

### A.14. Further Comparison to HIGGS

For a fairer comparison to the HIGGS method, in Tab. 19 compare it to SINQ with quantized auxiliaries (to ensure more similar memory usage).

### A.15. Additional Results on MoE Models

In Tab. 20 we show some perplexity results on MoE models to underline the flexibility of our method. These results further demonstrate that SINQ is able to outperform state-of-the-art calibration-free methods for weight quantization.

We additionally evaluate Qwen3-235B-A22B in thinking mode, on the AIME24 and AIME25 competition mathematics benchmarks (Avg@3, 16k-token generation budget). We select this model over DeepSeek-V2.5-236B because the latter, despite being MoE, is not trained as a reasoning model. SINQ and A-SINQ remain the best methods within their respective categories (Table 21).

### A.16. Weight Standard Deviation and Mean Absolute Inputs

We measure the predictive power of column-wise weight standard deviation on mean absolute inputs on further models in Tab. 22. A great variety of models (all we tested) show the relationship SINQ uses.

### A.17. Encoder Architectures

In Tab. 23 we report uniform, uncalibrated PTQ results on encoder-only and encoder-decoder architectures.

### A.18. Additional Baselines

We compare SINQ against three state-of-the-art calibration-based PTQ methods: OmniQuant (Shao et al.), FlexRound (Lee et al., 2023), and LRQ (Lee et al., 2025), a more recent extension of FlexRound. Despite being

*Table 14.* Accuracy (%) on HellaSwag, PIQA, and MMLU for Qwen3 models with 3-bit and 4-bit quantization. Higher is better.

| | | | Qwen3-14B | | | | Qwen3-32B | | |
|---|---|---|---|---|---|---|---|---|---|
| | | Method | *HellaSwag* | *PIQA* | *MMLU* | Avg. ↑ | *HellaSwag* | *PIQA* | *MMLU* | Avg. ↑ |
| | | Original (BF16) | *60.95* | *80.20* | *78.83* | 73.33 | *63.85* | *80.96* | *81.88* | 75.56 |
| CALIBRATION-FREE | 3-BIT | RTN | 56.99 | 77.80 | 75.01 | 69.93 | 46.98 | 71.82 | 78.53 | 65.78 |
| | | Hadamard + RTN | 49.66 | 73.45 | 67.53 | 63.55 | 50.43 | 75.41 | 78.10 | 67.98 |
| | | HQQ | 55.50 | **77.91** | 72.92 | 68.78 | 59.87 | 77.75 | 78.17 | 71.93 |
| | | **SINQ** *(ours)* | **58.03** | 77.20 | **75.82** | **70.35** | **60.65** | **79.49** | **78.89** | **73.11** |
| | 4-BIT | RTN | 60.11 | 79.11 | **78.44** | 72.55 | 61.22 | 78.78 | 81.78 | 73.93 |
| | | BnB (FP4) | 59.41 | 79.38 | 77.62 | 72.14 | 56.97 | 77.91 | 81.20 | 72.03 |
| | | BnB (NF4) | **60.47** | 79.71 | 78.23 | 72.80 | 63.12 | 79.98 | 81.60 | 74.90 |
| | | Hadamard + RTN | 58.45 | 78.67 | 76.58 | 71.23 | 62.90 | 78.94 | 80.99 | 74.28 |
| | | HQQ | 60.24 | 79.76 | 78.24 | 72.75 | 62.33 | 79.92 | **81.68** | 74.64 |
| | | **SINQ** *(ours)* | 60.05 | 79.54 | 78.00 | 72.53 | **63.20** | **80.85** | 81.63 | **75.23** |
| | | **SINQ (NF4)** *(ours)* | 60.35 | **79.72** | 78.37 | **72.81** | 63.18 | 80.52 | 81.32 | 75.00 |
| CALIBRATED | 3-BIT | GPTQ | **58.34** | 76.71 | 74.75 | 69.93 | 61.16 | 77.86 | 78.94 | 72.65 |
| | | Hadamard + GPTQ | 57.41 | **77.75** | 75.10 | 70.08 | 61.26 | 78.45 | 78.78 | 72.83 |
| | | **A-SINQ** *(ours)* | 58.16 | 77.15 | **75.40** | 70.24 | **61.47** | **79.22** | **79.00** | **73.23** |
| | 4-BIT | GPTQ | 60.55 | 79.43 | **78.11** | 72.70 | 63.22 | 80.20 | 81.36 | 74.93 |
| | | Hadamard + GPTQ | 60.27 | **79.60** | 77.85 | 72.57 | 63.01 | **81.01** | 80.98 | 75.00 |
| | | AWQ | 60.48 | 79.38 | 78.01 | 72.62 | **63.51** | 79.90 | 81.38 | 74.93 |
| | | **A-SINQ** *(ours)* | **60.84** | 79.22 | 78.07 | **72.71** | 63.43 | 80.03 | **81.61** | **75.02** |

*Table 15.* Weight-only PTQ on Phi models with 3-bit and 4-bit quantization, reporting perplexity and actual memory usage (GB). Lower is better for all metrics. In bold is the best result for a given setting.

| | | | Phi-2 (3B) | | | Phi-3 (4B) | | | Phi-4 (15B) | | |
|---|---|---|---|---|---|---|---|---|---|---|---|
| | | Method | Mem. | *Wiki2*↓ | *C4*↓ | Mem. | *Wiki2*↓ | *C4*↓ | Mem. | *Wiki2*↓ | *C4*↓ |
| | | Original (BF16) | 5.18 | *9.82* | *13.83* | 7.11 | *6.01* | *8.96* | 27.31 | *6.67* | *11.13* |
| CALIBRATION-FREE | 3-BIT | RTN | 1.57 | 12.24 | 16.27 | 1.96 | 9.74 | 12.39 | 7.82 | 7.29 | 12.23 |
| | | HQQ | 1.57 | 11.37 | 15.69 | 1.96 | 11.60 | 16.42 | 7.82 | 7.41 | 15.60 |
| | | **SINQ** *(ours)* | 1.64 | **11.07** | **15.23** | 1.99 | **9.56** | **12.14** | 7.91 | **7.28** | **12.19** |
| | 4-BIT | RTN | 1.81 | 10.30 | 14.40 | 2.28 | 6.95 | 9.71 | 9.18 | 6.64 | 11.38 |
| | | HQQ | 1.81 | 10.09 | 14.23 | 2.28 | 6.85 | 9.70 | 9.18 | 6.80 | 14.84 |
| | | **SINQ** *(ours)* | 1.86 | **9.98** | **14.09** | 2.29 | **6.79** | **9.68** | 9.32 | **6.61** | **11.32** |

*Table 18.* Single-token decoding (GEMV) latency comparison between QTIP W4A16 and SINQ W4A16 across matrix sizes $N$. QTIP results use the public precompiled kernels from https://github.com/Cornell-RelaxML/qtip for the supported shapes; SINQ uses an adapted W4A16 Marlin kernel extended to support the second scale. Latencies are reported in milliseconds (mean ± std over 200 timed runs after 50 warmup runs). Speedups are computed relative to FP16, with an additional column reporting SINQ vs. QTIP. Lower latency and higher speedup are better. The best result for a given setting is marked in **bold**.

| | FP16 | QTIP | | SINQ *(ours)* | | SINQ vs QTIP |
|---|---|---|---|---|---|---|
| $N$ | ms↓ | ms↓ | speedup↑ | ms↓ | speedup↑ | speedup↑ |
| 512 | 0.007 ± 0.005 | 0.035 ± 0.001 | 0.21× | **0.008 ± 0.001** | **0.93×** | **4.38×** |
| 1024 | 0.012 ± 0.001 | 0.036 ± 0.006 | 0.34× | **0.012 ± 0.001** | **1.02×** | **3.00×** |
| 2048 | 0.030 ± 0.002 | 0.039 ± 0.001 | 0.76× | **0.015 ± 0.001** | **1.97×** | **2.60×** |
| 3072 | 0.061 ± 0.001 | 0.046 ± 0.001 | 1.33× | **0.021 ± 0.001** | **2.84×** | **2.19×** |
| 4096 | 0.102 ± 0.001 | 0.056 ± 0.001 | 1.82× | **0.031 ± 0.001** | **3.32×** | **1.81×** |
| 6144 | 0.185 ± 0.001 | 0.091 ± 0.001 | 2.03× | **0.058 ± 0.001** | **3.16×** | **1.57×** |
| 8192 | 0.294 ± 0.000 | 0.131 ± 0.010 | 2.25× | **0.099 ± 0.002** | **2.98×** | **1.32×** |
| 12288 | 0.608 ± 0.001 | 0.220 ± 0.001 | 2.76× | **0.216 ± 0.001** | **2.81×** | **1.02×** |
| 16384 | 1.047 ± 0.001 | 0.381 ± 0.001 | 2.75× | **0.358 ± 0.003** | **2.92×** | **1.06×** |

*Table 19.* Comparison to HIGGS method with quantized auxiliary variables to better match the HIGGS memory use.

| | Method | Qwen3-1.7B | | | Qwen3-14B | | | Qwen3-32B | | |
|---|---|---|---|---|---|---|---|---|---|---|
| | | Mem. | *Wiki2*↓ | *C4*↓ | Mem. | *Wiki2*↓ | *C4*↓ | Mem. | *Wiki2*↓ | *C4*↓ |
| | Original (BF16) | 3.44 | *16.67* | *19.21* | 29.54 | *8.64* | *12.01* | 65.52 | *7.60* | *10.77* |
| 4-BIT | HIGGS (non-uniform) | 1.51 | 23.98 | 25.27 | 10.28 | 9.13 | 12.56 | 19.88 | 8.02 | 11.24 |
| | **SINQ** (NF4) *(ours)* | 1.42 | 16.94 | **19.83** | 10.56 | **8.72** | **12.13** | 20.73 | 7.83 | **10.97** |
| | **SINQ** (NF4) *(ours, q. aux.)* | **1.24** | **16.92** | 19.84 | **10.19** | **8.72** | **12.13** | **19.80** | **7.82** | 10.98 |

*Table 20.* Weight-only PTQ on **DeepSeek-V2-Lite** and **Qwen3-30B-A3B** MoE models with 3-bit and 4-bit quantization, reporting perplexity and actual memory usage (GB). Lower is better for all metrics. In bold is the best result for a given setting.

| | | DeepSeek-V2-Lite | | | Qwen3-30B-A3B | | |
|---|---|---|---|---|---|---|---|
| Setting | Method | Mem. | *Wiki2*↓ | *C4*↓ | Mem. | *Wiki2*↓ | *C4*↓ |
| Baseline | Original (BF16) | 32.55 | *6.31* | *8.83* | 61.06 | *8.70* | *12.15* |
| Calibration-free (3-bit) | RTN | 9.12 | 7.94 | 10.98 | 15.10 | 12.28 | 15.89 |
| | HQQ | 9.12 | 8.36 | 11.74 | 15.10 | 10.52 | 14.39 |
| | **SINQ** *(ours)* | 9.02 | **7.45** | **10.32** | 15.13 | **10.19** | **13.62** |
| Calibration-free (4-bit) | RTN | 10.63 | 6.59 | 9.19 | 18.07 | 9.04 | 12.64 |
| | BnB | 10.63 | 6.82 | 9.49 | 18.08 | 9.68 | 12.93 |
| | HQQ | 10.85 | 6.61 | 9.18 | 18.07 | 9.14 | 12.64 |
| | **SINQ** *(ours)* | 10.50 | **6.49** | **9.07** | 18.13 | **9.02** | **12.41** |

*Table 21.* Downstream evaluation of 4-bit quantization on Qwen3-235B-A22B (MoE, thinking mode) on competition mathematics benchmarks (AIME24, AIME25), reporting Avg@3. We compare calibration-free methods (RTN, SINQ) and calibrated methods (AWQ, ASINQ) against the FP16 baseline. Higher is better. The best result among methods of the same type (calibration-free vs. calibrated) is marked in **bold**. Generation budget: 16k tokens.

| | Method | *AIME24*↑ | *AIME25*↑ |
|---|---|---|---|
| | Original (FP16) | *78.90* | *75.60* |
| NO CAL. | RTN | 64.40 | 57.80 |
| | **SINQ** *(ours)* | **73.30** | **70.00** |
| CAL. | AWQ | 74.40 | 67.80 |
| | **A-SINQ** *(ours)* | **76.70** | **70.00** |

*Table 22.* Mean $R^2$ between $\mu_x$ (mean absolute input) and three reference vectors, computed per layer and then averaged across layers. $\mu_x\sigma_W\text{-}R^2$: correlation with $\sigma_W$, the per-input-channel standard deviation of the weight matrix taken over the output dimension. **shuffled-**$\sigma_W\text{-}R^2$: same as $\mu_x\sigma_W\text{-}R^2$ but with $\sigma_W$ randomly permuted across channels, providing a chance-level lower bound. $\mu_x\mu_x\text{-}R^2$: correlation between $\mu_x$ computed on two different calibration datasets, providing an upper bound on how much input-channel variance any calibration data could in principle explain. Across all model families, $\mu_x\sigma_W\text{-}R^2$ remains close to the shuffled baseline and far below the cross-dataset ceiling, indicating that weight statistics carry little information about activation-channel magnitudes.

| Model | Type | $\mu_x\sigma_W\text{-}R^2$ | shuffled-$\sigma_W\text{-}R^2$ | $\mu_x\mu_x\text{-}R^2$ |
|---|---|---|---|---|
| Qwen3-0.6B | LLM | 0.2607 | 0.0004 | 0.5839 |
| Qwen3-1.7B | LLM | 0.1825 | 0.0003 | 0.6191 |
| Qwen3-4B | LLM | 0.1617 | 0.0002 | 0.6644 |
| Qwen3-8B | LLM | 0.1595 | 0.0008 | 0.6994 |
| Qwen3-14B | LLM | 0.1660 | 0.0001 | 0.7146 |
| Qwen3-32B | LLM | 0.1990 | 0.0002 | 0.7002 |
| Llama-2-13B | LLM | 0.2058 | 0.0002 | 0.5434 |
| Llama-3.1-8B | LLM | 0.2324 | 0.0002 | 0.6583 |
| Llama-3.2-3B | LLM | 0.2321 | 0.0003 | 0.7128 |
| Mamba-2.8B | SSM | 0.1989 | 0.0021 | 0.6054 |
| FalconH1-1.5B | Hybrid-LLM | 0.2123 | 0.0006 | 0.5949 |
| FalconH1-3B | Hybrid-LLM | 0.1979 | 0.0003 | 0.5445 |
| Swin-B | Vision | 0.1983 | 0.0021 | 0.8023 |
| SwinV2-B | Vision | 0.1908 | 0.0027 | 0.8034 |
| ESM2-8M | Protein | 0.3631 | 0.0021 | 0.9457 |
| ESM2-35M | Protein | 0.3894 | 0.0124 | 0.9217 |

*Table 23.* Weight-only uncalibrated uniform PTQ on encoder-only (BERT, RoBERTa) and encoder-decoder (Flan-T5) models, demonstrating the architecture-agnosticism of SINQ. We report accuracy (↑) for classification tasks (SST-2, QNLI) and ROUGE-2 (↑) for summarization (XSum). Higher is better for all metrics. The best result for a given setting is marked in **bold**.

| | Model | Task | FP16 | RTN | HQQ | **SINQ** *(ours)* |
|---|---|---|---|---|---|---|
| **4-BIT** | BERT-base | SST-2 ↑ | *92.43* | 91.74 | 91.97 | **92.05** |
| | BERT-base | QNLI ↑ | *91.56* | 91.07 | 91.18 | **91.27** |
| | RoBERTa-base | SST-2 ↑ | *94.04* | 93.12 | **94.38** | 94.01 |
| | RoBERTa-base | QNLI ↑ | *92.64* | 88.49 | 91.58 | **92.53** |
| | Flan-T5-base | XSum (R2) ↑ | *10.72* | 10.23 | 10.62 | **10.65** |
| | Flan-T5-large | XSum (R2) ↑ | *15.52* | 13.01 | 13.79 | **13.80** |
| **3-BIT** | BERT-base | SST-2 ↑ | *92.43* | 90.71 | 91.40 | **91.63** |
| | BERT-base | QNLI ↑ | *91.56* | 90.43 | 90.50 | **90.63** |
| | RoBERTa-base | SST-2 ↑ | *94.04* | 86.35 | 88.88 | **93.23** |
| | RoBERTa-base | QNLI ↑ | *92.64* | 60.64 | 59.80 | **88.27** |
| | Flan-T5-base | XSum (R2) ↑ | *10.72* | 8.46 | 8.79 | **10.70** |
| | Flan-T5-large | XSum (R2) ↑ | *15.52* | 5.95 | 9.49 | **13.59** |

calibration-free, SINQ outperforms all three baselines on Llama-2-7B, -13B, and -70B at both 3-bit and 4-bit precision (Table 24). OmniQuant, FlexRound, and LRQ numbers are taken from Table 7 of the LRQ paper.

### A.19. Decode-Dominant Tasks

A common concern with quantized LLMs is degraded performance on downstream reasoning tasks. We evaluate SINQ and its calibrated variant A-SINQ on IFEval (prompt-level strict instruction-following) and GSM8K-CoT (4-shot chain-of-thought math), comparing against RTN among calibration-free methods and AWQ among calibrated methods. Comparisons are restricted to methods with kernels available in fast serving frameworks such as vLLM. SINQ outperforms RTN across all settings — most strikingly on GSM8K with Qwen3-1.7B, where SINQ recovers 17.7 points over RTN (63.80 vs. 46.10) — and A-SINQ matches or exceeds AWQ on every benchmark and model (Table 25).

### A.20. Single Layer Weight-Standard-Deviation Stability

The derivation in Eq. 4 is intended as an intuition; the real test of the idea lies in end-to-end performance of actual deep models. Nonetheless, we run additional experiments to confirm that the predicted log-log slope of $-0.500$ holds across a range of optimizers and target distributions. Results are reported in Tables 26 and 27: across all five optimizers and all five distributions tested, the fitted slope remains close to the theoretical value, with high $R^2$. The two cases with slightly weaker fits — Adagrad and Muon — still recover slopes within 10% of the predicted value.

### A.21. Comparison to Kurtosis Local Proxy

Kurtail (Akhondzadeh et al.) suggest the kurtosis as a local proxy metric and optimization target for making matrices more easily quantizable, in the context of finding optimal rotations to apply to each layer. We find that 1) our imbalance optimization substantially reduces the average kurtosis of both rows and columns and 2) that directly minimizing kurtosis (while increasing imbalance) in our setting decreases end-to-end accuracy, see Fig. 10. This indicates that *for the dual-scaling setting*, imbalance is a better proxy target for ease of quantization than kurtosis.

*Table 24.* Comparison of SINQ against state-of-the-art calibration-based PTQ methods (OmniQuant, FlexRound, LRQ) on Llama-2 models, reporting WikiText-2 perplexity. Lower is better. The best result for a given setting is marked in **bold**. Although SINQ is calibration-free, it outperforms all three calibrated baselines across model sizes and bit-widths.

| | Method | *Llama-2-7B↓* | *Llama-2-13B↓* | *Llama-2-70B↓* |
|---|---|---|---|---|
| | Original (FP16) | 5.47* | 4.88 | 3.31 |
| 3-BIT | OmniQuant[‡] | 6.57 | 5.58 | 3.93 |
| | FlexRound[‡] | 6.34 | 5.59 | 3.92 |
| | LRQ[‡] | 6.48 | 5.57 | 3.89 |
| | **SINQ** *(ours)* | **6.14** | **5.22** | **3.76** |
| 4-BIT | OmniQuant[‡] | 5.74 | 5.02 | 3.47 |
| | FlexRound[‡] | 5.83 | 5.01 | 3.45 |
| | LRQ[‡] | 5.75 | 5.02 | 3.47 |
| | **SINQ** *(ours)* | **5.60** | **4.96** | **3.42** |

[‡] Numbers from Table 7 of LRQ (Lee et al., 2025).

*Table 25.* Downstream evaluation of 4-bit quantization on instruction-following (IFEval, prompt-level strict accuracy) and mathematical reasoning (GSM8K-CoT, 4-shot accuracy). We compare calibration-free methods (RTN, SINQ) and calibrated methods (AWQ, ASINQ) against the FP16 baseline. Higher is better. The best result among methods of the same type (calibration-free vs. calibrated) is marked in **bold**. Sampling: temperature=0.7, top-$p$=0.8, top-$k$=20, presence penalty=1.5.

| | | **IFEval ↑** | | |
|---|---|---|---|---|
| | Method | *Qwen3-1.7B* | *Qwen3-4B* | *Qwen3-14B* |
| | Original (FP16) | 67.28 | 78.93 | 85.77 |
| NO CAL. | RTN | 60.26 | 75.23 | 84.10 |
| | **SINQ** *(ours)* | **63.21** | **77.08** | **85.77** |
| CAL. | AWQ | 64.69 | 77.45 | 85.03 |
| | **A-SINQ** *(ours)* | **66.17** | **77.82** | **85.21** |
| | | **GSM8K-CoT ↑** | | |
| | Method | *Qwen3-1.7B* | *Qwen3-4B* | *Qwen3-14B* |
| | Original (FP16) | 72.78 | 78.47 | 86.73 |
| NO CAL. | RTN | 46.10 | 72.86 | 81.47 |
| | **SINQ** *(ours)* | **63.80** | **75.96** | **84.08** |
| CAL. | AWQ | 66.19 | 76.57 | 85.11 |
| | **A-SINQ** *(ours)* | **66.31** | **76.82** | **86.31** |

*Table 26.* Log-log fit of the per-channel weight standard deviation versus the corresponding pre-activation magnitude, for a single linear layer trained on an i.i.d. Gaussian target under different optimizers. Reported are the fit $R^2$ and the fitted slope; the expected slope under the theoretical relationship is $-0.500$.

| **Optimizer** | $R^2 \uparrow$ | slope |
|---|---|---|
| Adam | 0.993 | $-0.500$ |
| AdamW | 0.993 | $-0.499$ |
| Adagrad | 0.896 | $-0.448$ |
| Muon | 0.954 | $-0.458$ |
| RMSProp | 0.994 | $-0.497$ |

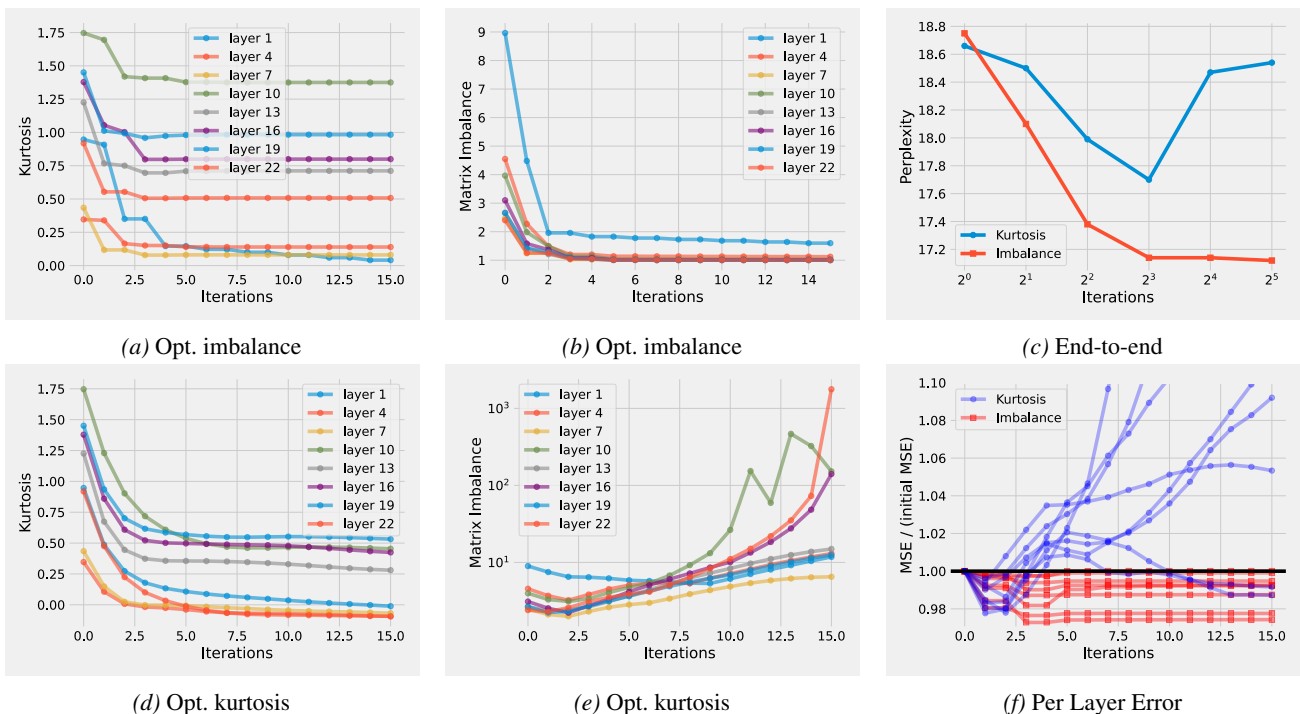

*Figure 10.* Results on Qwen3-1.7B. Minimizing the imbalance with our algorithm (a and b) decreases both the imbalance and the kurtosis. Minimizing the kurtosis directly with gradient descent (d and e) yields lower kurtosis, but causes a large imbalance; note the log-scale on (d). Finally (c) and (f) show the end-to-end perplexity on wikitext2 and per-layer RTN MSE improvement when optimizing imbalance or kurtosis, respectively.

*Table 27.* Same log-log fit as Table 26, varying the target distribution while keeping the optimizer fixed to Adam. The expected slope under the theoretical relationship is $-0.500$.

| Distribution | $R^2 \uparrow$ | slope |
|---|---|---|
| Gaussian | 0.993 | $-0.504$ |
| Uniform | 0.995 | $-0.507$ |
| Log-normal | 0.993 | $-0.505$ |
| MoG | 0.992 | $-0.503$ |
| Laplace | 0.992 | $-0.501$ |

