# OpenReview forum: "SINQ: Sinkhorn-Normalized Quantization for Calibration-Free Low-Precision LLM Weights"
_ICML.cc/2026/Conference — ICML 2026 regular_

### Official Review · Reviewer_WMPW · 2026-03-08

**Soundness:** 4
**Presentation:** 4
**Significance:** 3
**Originality:** 4
**Overall Recommendation:** 5
**Confidence:** 4

**Summary:**

This paper proposes SINQ, a data-free weight quantization method for LLMs that applies scaling along both rows and columns of linear layer weights. The key heuristic is based on an interesting observation on weight column std and activation abs mean. Based on this, a data-free algorithm for the second dim scales is proposed. Extensive experiments are conducted to test the heuristics and the SINQ quantization.

**Compliance With Llm Reviewing Policy:**

Affirmed.

**Final Justification:**

The method proposed in this paper is novel and well-tested through evaluation

The author addressed my concerns through rebuttal

I kept my positive score

**Key Questions For Authors:**

- Could the author elaborate the the blue bars and gray bars in Figure 2? They are $R^2$-value for testing the heuristic of the reciprocal column-wise weight std. and the average input abs, but what does Alignment input and cross-dataset mean here? How are the $R^2$ values calculated exactly for this experiment?
- The observation in Figure 3 is very interesting.
  - Could the author explain this observation?
  - Is this quantizer-dependent? Say If the quantizer is MXFP4, instead of RTN
  - Does this observation still hold for larger models?  (Fig.3 is Qwen3-1.7B)

**Limitations:**

Please refer to the weakness and questions above. Besides these, I do not see other limitations that the authors need to discuss

**Strengths And Weaknesses:**

## Strength

- This paper is clear and easy-to-follow.
- The observation on weight column std and activation abs mean is interesting and enlightening, especially the comparison with rotation-based methods. This may guide followup works in the field of weight quantization for LLMs
- Extensive experiments were conducted. In additional to 'conventional' evaluation on LLM perplexity, this paper also reports flip rates, a metric for evaluation quantized models more reliable than accuracy. Besides, the ablation study and test of heuristics in Fig 2, 3 are helpful for understanding the advantage of SINQ's dual scaling.
- SINQ achieves impressive model performance in terms of data-free element quantization for LLMs. I regard this as valuable work though many weight quantization works shift to vector quantization.

## Weakness

I didn't see obvious weakness of this work. Here are minor ones

- It might be clearer if a linear layer formula (Eq. 7) adopted in this paper is put at the beginning, otherwise the reader need to infer/guess the form taken by this paper: $y = xW$, $y = Wx$, or $y = xW^T$
- Recent vector quantization based methods usually achieves better model performance. I'm curious about a comparison on inference latency/throughput - model performance between SINQ and SoTA vector quantization methods. Note that I do not think this is required, but a comparison may help reader understand the application scenarios/positioning of SINQ.

---

> ### Author Rebuttal · Authors · 2026-03-31
>
> Thank you for your helpful feedback.
>
> ## Questions
>
> 1. "Could the author elaborate the the blue bars and gray bars in Figure 2? [...]"
>
> Thank you for pointing this out, we will clarify in the paper how exactly this is computed. R^2 is computed as the square of the pearson-correlation (as implemented e.g. by scipy) between mu_x the mean (over batch + sequence dimension) of the absolute value of the input x to the layer and W_s the standard deviation of the weight matrix over the output dimension (note: both these quantities are vectors whose length is the input dimension of the layer). The R^2 over this input dimension tells us how much of the variance across input channels in the mean abs. input can be explained by the standard deviation in W (XW-R^2 in the table below).
> For the grey bars we compute the same R^2 but compare mu_x for different datasets 1) python code 2) chinese wikipedia phrases (xx-R^2 in the table).
> For the red bars we shuffle W_s before computing the variance to be sure the correlation is not spurious, but relates to the ordering of the channels (shuffled... in the table).
> The red bars indicate a reasonable lower bound: The explained variance one might observe by chance. The grey bars are a reasonable upper bound: The explained variance between different hypothetical calibration data.
>
> We further confirmed that this observation is stable across a wider pool of models, including two vision and protein models. In following table we report the mean of the R^2 for the different conditions.
>
> | Model | Type | XW-R^2 | shuffled-Wstd-R^2 | xx-R^2 |
> | :--- | :--- | :--- | :--- | :--- |
> | Qwen3-0.6B | LLM | 0.2607 | 0.0004 | 0.5839 |
> | Qwen3-1.7B | LLM | 0.1825 | 0.0003 | 0.6191 |
> | Qwen3-4B | LLM | 0.1617 | 0.0002 | 0.6644 |
> | Qwen3-8B | LLM | 0.1595 | 0.0008 | 0.6994 |
> | Qwen3-14B | LLM | 0.1660 | 0.0001 | 0.7146 |
> | Qwen3-32B | LLM | 0.1990 | 0.0002 | 0.7002 |
> | Llama-2-13B | LLM | 0.2058 | 0.0002 | 0.5434 |
> | Llama-3.1-8B | LLM | 0.2324 | 0.0002 | 0.6583 |
> | Llama-3.2-3B | LLM | 0.2321 | 0.0003 | 0.7128 |
> | Mamba-2.8B | SSM | 0.1989 | 0.0021 | 0.6054 |
> | FalconH1-1.5B | Hybrid-LLM | 0.2123 | 0.0006 | 0.5949 |
> | FalconH1-3B | Hybrid-LLM | 0.1979 | 0.0003 | 0.5445 |
> | Swin-B | Vision | 0.1983 | 0.0021 | 0.8023 |
> | SwinV2-B | Vision | 0.1908 | 0.0027 | 0.8034 |
> | ESM2-8M | Protein | 0.3631 | 0.0021 | 0.9457 |
> | ESM2-35M | Protein | 0.3894 | 0.0124 | 0.9217 |
>
> 2. "The observation in Figure 3 is very interesting. Could the author explain this observation? [...] Does this observation still hold for larger models? (Fig.3 is Qwen3-1.7B)"
>
> Hadamard rotation by desgin decreases the matrix reconstruction error, but as we know from calibrated approaches, the best quantizations are NOT the ones with the lowest matrix reconstruction error: Errors in some positions matter more than others, because the input data is not random uniform. In contrast SINQ does not aim to decrease the matrix reconstruction error and sometimes also increases it; instead the aim is to rescale the matrix so that RTN quantization will mostly cause errors in places where it does not matter (similar to calibrated methods).
>
> Regarding MXFP4 and larger models, we expect the above to hold as well. To confirm this, we ran the same experiment as in Figure 3, but with larger Qwen3 models and with MXFP4 quantization in addtion to integer. The conclusion is that also in these settings Hadamard rotation is consistently better at reconstructing weight matrices, but SINQ usually achieves better output accuracy. The measurements are summarized in this table.
>
> | Model | Format | ΔMatSINQ | ΔMatHad | ΔActSINQ | ΔActHad |
> | :--- | :--- | :--- | :--- | :--- | :--- |
> | 1.7B | INT | +0.000001 | -0.000510 | -0.009493 | +0.090778 |
> | 1.7B | MXFP4 | +0.000000 | -0.000071 | -0.001951 | +0.062639 |
> | 14B | INT | -0.000002 | -0.000325 | -0.000228 | +0.021058 |
> | 14B | MXFP4 | +0.000006 | -0.000041 | -0.005413 | -0.004898 |
> | 32B | INT | -0.000003 | -0.000293 | -0.000040 | +0.008226 |
> | 32B | MXFP4 | +0.000006 | -0.000035 | -0.003115 | -0.009607 |
>
> *(Δ = Method - RTN; negative means better than RTN)*
>
> ### Weaknesses
> 1. "It might be clearer if a linear layer formula [...] is put at the beginning [...]"
>
> Thank you, we will mention the definition we use for the linear layer earlier. We chose this form, because it aligns with pytorch code.

---

> > ### Author Rebuttal · Reviewer_WMPW · 2026-04-02
> >
> > The rebuttal has addressed my concerns, so I keep the positive score.

---

> > > ### Author Response · Authors · 2026-04-07
> > >
> > > Thank you for your time and for noting that all concerns have been resolved.
> > >
> > > We did notice you were also curious about this:
> > >
> > > > "I'm curious about a comparison on inference latency/throughput - model performance between SINQ and SoTA vector quantization methods. Note that I do not think this is required, but a comparison may help reader understand the application scenarios/positioning of SINQ"
> > >
> > > Since this comparison was raised as an interesting but non-mandatory point, we did not have results ready while writing the first rebuttal comment, but we have since had time to run an additional latency experiment.
> > >
> > > We chose QTIP as a representative Vector-Quantization (VQ) baseline because it provides publicly available 4-bit inference kernels, making it one of the few recent VQ methods for which such a comparison is currently possible. We compared **QTIP W4A16** and **SINQ W4A16** in a single-token decoding (GEMV) setting (the one for which kernels are available), using the public QTIP precompiled kernels (https://github.com/Cornell-RelaxML/qtip) for the supported shapes. For SINQ, we used an adapted W4A16 Marlin kernel extended to support the second scale. Results are computed over 200 timed runs (after 50 warmup runs). We report speedup over FP16 (higher is better) and SINQ vs QTIP speedup (higher is better):
> > >
> > > | N | FP16 ms | QTIP ms | QTIP speedup | SINQ ms | SINQ speedup | SINQ vs QTIP speedup|
> > > |---:|---:|---:|:---:|---:|:---:|:---:|
> > > | 512 | 0.007 ± 0.005 | 0.035 ± 0.001 | 0.21× | 0.008 ± 0.001 | 0.93× | **4.38×** |
> > > | 1024 | 0.012 ± 0.001 | 0.036 ± 0.006 | 0.34× | 0.012 ± 0.001 | 1.02× | **3.00×** |
> > > | 2048 | 0.030 ± 0.002 | 0.039 ± 0.001 | 0.76× | 0.015 ± 0.001 | 1.97× | **2.60×** |
> > > | 3072 | 0.061 ± 0.001 | 0.046 ± 0.001 | 1.33× | 0.021 ± 0.001 | 2.84× | **2.19×** |
> > > | 4096 | 0.102 ± 0.001 | 0.056 ± 0.001 | 1.82× | 0.031 ± 0.001 | 3.32× | **1.81×** |
> > > | 6144 | 0.185 ± 0.001 | 0.091 ± 0.001 | 2.03× | 0.058 ± 0.001 | 3.16× | **1.57×** |
> > > | 8192 | 0.294 ± 0.000 | 0.131 ± 0.010 | 2.25× | 0.099 ± 0.002 | 2.98× | **1.32×** |
> > > | 12288 | 0.608 ± 0.001 | 0.220 ± 0.001 | 2.76× | 0.216 ± 0.001 | 2.81× | **1.02×** |
> > > | 16384 | 1.047 ± 0.001 | 0.381 ± 0.001 | 2.75× | 0.358 ± 0.003 | 2.92× | **1.06×** |
> > >
> > > This suggests that SINQ, and more generally standard W4A16 INT4 methods with mature kernels, appear to offer a better latency profile than the tested vector-quantization baseline, especially at small-to-medium sizes, while the gap narrows for larger matrices (which is consistent with the regime becoming less memory-bound).
> > >
> > > We do agree that this comparison is useful for clarifying the practical positioning of SINQ, and we will include this discussion in the final version.

---

### Official Review · Reviewer_L3dA · 2026-03-13

**Soundness:** 3
**Presentation:** 3
**Significance:** 3
**Originality:** 3
**Overall Recommendation:** 4
**Confidence:** 3

**Summary:**

SINQ proposes a dual-scale PTQ parameterization (one scale per row, one per column) combined with a Sinkhorn-Knopp-inspired iterative algorithm that normalizes both row and column standard deviations before quantization. The key insight is that column-wise weight standard deviations are predictive of input activation magnitudes, enabling calibration-free pseudo-activation-aware quantization. Evaluated primarily on the Qwen3 family, SINQ consistently outperforms RTN, HQQ, and Hadamard+RTN across 3-bit and 4-bit settings, with negligible inference and quantization overhead.

**Compliance With Llm Reviewing Policy:**

Affirmed.

**Key Questions For Authors:**

The imbalance criterion I(W) is used as the optimization target in Algorithm 1, but the ultimate goal is to minimize perplexity. Is there a theoretical or empirical justification for why minimizing I(W) is a good proxy? An ablation comparing I(W) against alternative objectives such as minimizing output activation MSE directly would strengthen this.

**Limitations:**

Yes

**Strengths And Weaknesses:**

The core observation that column-wise weight std. dev. correlates with input activation scale is empirically well-supported across multiple model families and theoretically grounded for the Adam/Gaussian toy setting. This is a genuine and useful insight.

Results are comprehensive: Qwen3 (0.6B–235B), Llama, Phi, DeepSeek, MoE models, GGUF integration, reasoning benchmarks, flip-rate metrics, and quantization/inference timing are all covered.

The no-overhead absorption of the second scale into preceding layers is a practically important contribution, and the GGUF integration results demonstrate real-world applicability.

The comparison of matrix vs. activation reconstruction error between SINQ and Hadamard rotation is a strong and clarifying result that cleanly motivates the method over rotation-based alternatives.

The theoretical justification (Eq. 4) is only derived for a single linear layer trained with Adam on an i.i.d. Gaussian target. The paper does not discuss how well this holds under different optimizers, non-Gaussian data, or multi-layer interactions.

The imbalance metric I(W) as the optimization target is a somewhat ad hoc design choice. No ablation or justification is provided for why this objective is preferable to alternatives such as directly minimizing output activation MSE.

Experiments cover only decoder-only transformers. The architecture-agnosticism claim is not validated on encoder-based or cross-attention architectures.

QuIP# and QTIP comparisons appear only in a separate appendix on Llama2 with a different context length, preventing direct apples-to-apples comparison with the main results.

---

> ### Author Rebuttal · Authors · 2026-03-31
>
> Thank you for your helpful feedback.
>
> ## Questions
> 1. "The imbalance criterion I(W) is used as the optimization target in Algorithm 1, but the ultimate goal is to minimize perplexity [...] An ablation comparing I(W) against alternative objectives such as minimizing output activation MSE directly would strengthen this." + "The imbalance metric I(W) as the optimization target is a somewhat ad hoc design choice. No ablation or justification is provided for why this objective is preferable to alternatives such as directly minimizing output activation MSE."
>
> In our uncalibrated setting, output activation MSE cannot be computed, because inputs are unknown.
> We instead tried an alternate proxy metric for direct optimization that we will add to the paper: The kurtosis (as proposed in the 'Kurtail' paper, https://arxiv.org/abs/2503.01483). We find that 1) both optimizing for imbalance and for kurtosis decreases the kurtosis 2) optimizing for kurtosis directly leads to a lower kurtosis than optimizing for imbalance 3) optimizing for kurtosis often increases the imbalance 4) optimizing for imbalance leads to lower end-to-end error. The reason for this is, that the kurtosis has no sense of typical input scales (in contrast to the imbalance).
>
> ## Weaknesses
>
> 1. "The theoretical justification (Eq. 4) is only derived for a single linear layer trained with Adam on an i.i.d. Gaussian target. The paper does not discuss how well this holds under different optimizers [...]"
>
> Note that this is just an intuition, the real test of the idea lies in the end-to-end performance of actual deep models. Still, we run additional experiments to confirm this relationship for different optimizers and distributions and will add them to the paper for better grounding of this motivation. The expected log-log slope for $s_W \propto 1/\sqrt{s_x}$ is -0.500 for the same relationship to hold.
>
> | Optimizer | $R^2$ | log-log slope |
> | :--- | :--- |  :--- |
> | Adam | 0.993 |  -0.500 |
> | AdamW | 0.993 | -0.499 |
> | Adagrad | 0.896 | -0.448 |
> | Muon | 0.954  | -0.458 |
> | RMSProp | 0.994 | -0.497 |
>
>
> And with different distributions:
>
> | Distribution | $R^2$ | log-log slope |
> | :--- | :--- | :--- |
> | Gaussian | 0.993 | -0.504 |
> | Uniform | 0.995 |  -0.507 |
> | Log-normal | 0.993  | -0.505 |
> | MoG | 0.992 | -0.503 |
> | Laplace | 0.992 | -0.501 |
>
> 2. "Experiments cover only decoder-only transformers. The architecture-agnosticism claim is not validated on encoder-based or cross-attention architectures."
>
> We add comparisons on encoder-decoder and encoder-only models to further demonstrate the architecture-agnosticism of SINQ
>
> ### 4-bit (Acc.)
>
> | Model          | Task   | FP16  | RTN   | HQQ   | SINQ  |
> |:---------------|:-------|------:|------:|------:|------:|
> | BERT-base      | SST-2 ↑ | 92.43 | 91.74 | 91.97 | **92.05** |
> | BERT-base      | QNLI ↑  | 91.56 | 91.07 | 91.18 | **91.27** |
> | RoBERTa-base   | SST-2 ↑ | 94.04 | 93.12 | 94.38 | **94.01** |
> | RoBERTa-base   | QNLI ↑  | 92.64 | 88.49 | 91.58 | **92.53** |
>
> ### 4-bit (Rouge2 ↑)
> | Model         | Task | FP16  | RTN   | HQQ   | SINQ   |
> |:--------------|:-----|------:|------:|------:|-------:|
> | Flan-T5-base  | XSum | 10.72 | 10.23 | 10.62 | **10.65**  |
> | Flan-T5-large | XSum | 15.52 | 13.01 | 13.79 | **13.80**  |
>
> ### 3-bit (Acc.)
>
> | Model          | Task   | FP16  | RTN   | HQQ   | SINQ  |
> |:---------------|:-------|------:|------:|------:|------:|
> | BERT-base      | SST-2 ↑ | 92.43 | 90.71 | 91.40 | **91.63** |
> | BERT-base      | QNLI ↑  | 91.56 | 90.43 | 90.50 | **90.63** |
> | RoBERTa-base   | SST-2 ↑ | 94.04 | 86.35 | 88.88 | **93.23** |
> | RoBERTa-base   | QNLI ↑  | 92.64 | 60.64 | 59.80 | **88.27** |
>
> ### 3-bit (Rouge2 ↑)
> | Model         | Task | FP16  | RTN  | HQQ  | SINQ  |
> |:--------------|:-----|------:|-----:|-----:|------:|
> | Flan-T5-base  | XSum | 10.72 | 8.46 | 8.79 | **10.70** |
> | Flan-T5-large | XSum | 15.52 | 5.95 | 9.49 | **13.59** |
>
> 3. "QuIP# and QTIP comparisons appear only in a separate appendix on Llama2 with a different context length, preventing direct apples-to-apples comparison with the main results."
>
> This is an interesting point and we do not know why the QTIP authors chose a non-standard context length (e.g., AWQ, GPTQ, QuaRot papers share the same context length for ppl). That is why we reproduced our results in their setting for the fairest possible comparison.
>
> Note that QUIP# and QTIP rely on codebooks. Differently from SINQ these methods are incompatible with activation quantization and require non-standard operations/kernels which are, as far as we know, not available, for example, on NPUs and TPUs.

---

### Official Review · Reviewer_cLRS · 2026-03-13

**Soundness:** 3
**Presentation:** 3
**Significance:** 3
**Originality:** 3
**Overall Recommendation:** 5
**Confidence:** 4

**Summary:**

The authors observe that column-wise weight matrix variances are predictive of the typical activation magnitudes, which makes calibration-free pseudo-activation-aware quantization possible. They propose SINQ to add the second-axis scale factor based on a fast
Sinkhorn–Knopp–style algorithm that finds column-wise scales that mimic activation-awareness, while preventing row-wise kurtosis from increasing.

**Compliance With Llm Reviewing Policy:**

Affirmed.

**Final Justification:**

As all of my concerns are addressed, I lean toward acceptance.

**Key Questions For Authors:**

- In Table 6, why is W4A16 SINQ compared with W16A16 AWQ, not W4A16 AWQ?

**Strengths And Weaknesses:**

**Strengths**

- The authors presents an interesting idea (i.e., calibration-free pseudo-activation-aware quantization) and it seems working well.

- They conduct extensive experiments across different model scales.

**Weaknesses**

- Any generation benchmark score (e.g., IFEval, GSM8k) is not reported, which makes it hard to determine whether SINQ is effective for LLMs or not. It would be more instructive if there are also generation benchmark scores across different model scales.

- Although SINQ is calibration-free, it would be necessary to compare SINQ with other state-of-the-art PTQ methods (e.g., FlexRound [1], OmniQuant [2], EfficientQAT [3]) to further validate the efficacy of SINQ.

- In Table 13, the authors show the experimental results of MoE, which is impressive. However, given that the main benchmarks for FP MoEs are generally reasoning tasks, it would be required to measure AIME24 and/or AIME25 scores in order to highlight that SINQ is also effective for MoE models.

[1] FlexRound: Learnable Rounding based on Element-wise Division for Post-Training Quantization, ICML 2023

[2] OmniQuant: Omnidirectionally Calibrated Quantization for Large Language Models, ICLR 2024

[3] EfficientQAT: Efficient Quantization-Aware Training for Large Language Models, ACL 2025

---

> ### Author Rebuttal · Authors · 2026-03-31
>
> Thank you for your helpful review.
>
> ## Questions
> Both AWQ and SINQ run at W4A16. We now realize that the caption is ambiguous in the paper and we will reformulate it. In the same setting (W4A16) AWQ and SINQ provide about the same end-to-end speed-up over W16A16.
>
> ## Weaknesses
> 1. "Any generation benchmark score (e.g., IFEval, GSM8k) is not reported"
>
> We do agree with the reviewer that there are cases where quantized LLMs do not generalize well, especially to reasoning tasks. Unfortunately, benchmarking on these is computationally intensive, and we can only compare against methods that provide kernels for fast serving frameworks (e.g., vLLM).
>
> Also in this setting SINQ performs well.
>
> Setting: temperature=0.7,top_p=0.8,top_k=20,presence_penalty=1.5
>
> ### IFEval (prompt level, strict ↑):
>
> | Method | Qwen3-1.7B | Qwen3-4B | Qwen3-14B | Llama2-7B |
> |:-------|-----------:|---------:|----------:|----------:|
> | FP16   | 67.28      | 78.93    | 85.77     | 34.20     |
> | *Calibration-free* |  |  |  |  |
> | RTN    | 60.26      | 75.23    | 84.10     | 30.68     |
> | **SINQ** | **63.21** | **77.08** | **85.77** | **33.09** |
> | *Calibrated* |  |  |  |  |
> | AWQ    | 64.69      | 77.45    | 85.03     | 33.09     |
> | **ASINQ** | **66.17** | **77.82** | **85.21** | **33.64** |
>
>
> ### GSM8k-CoT (fewshot 4 ↑)
>
> | Method | Qwen3-1.7B | Qwen3-4B | Qwen3-14B |
> |:-------|-----------:|---------:|----------:|
> | FP16   | 72.78      | 78.47    | 86.73     |
> | *Calibration-free* |  |  |  |
> | RTN    | 46.10      | 72.86    | 81.47     |
> | **SINQ** | **63.80** | **75.96** | **84.08** |
> | *Calibrated* |  |  |  |
> | AWQ    | 66.19      | 76.57    | 86.11     |
> | **ASINQ** | **66.31** | **76.82** | **86.31** |
>
> 2. "Although SINQ is calibration-free, it would be necessary to compare SINQ with other state-of-the-art PTQ methods (e.g., FlexRound [1], OmniQuant [2], EfficientQAT [3]) to further validate the efficacy of SINQ."
>
> Thank you for these additional references. We will add them to the paper (we also add a comparison with a newer version of FlexRound we did not know called LRQ [4]). We find that SINQ outperforms FlexRound, OmniQuant and LRQ.
>
> Wiki ppl:
>
> | Method | Llama-2-7B | Llama-2-13B | Llama-2-70B |
> |:-------|-----------:|------------:|------------:|
> | FP16 | 5.47* | 4.88 | 3.31 |
> | **3 bit** |  |  |  |
> | OmniQuant | 6.57 | 5.58 | 3.93 |
> | FlexRound | 6.34 | 5.59 | 3.92 |
> | LRQ | 6.48 | 5.57 | 3.89 |
> | **SINQ** | **6.14*** | **5.22** | **3.76** |
> | **4 bit** |  |  |  |
> | OmniQuant | 5.74 | 5.02 | 3.47 |
> | FlexRound | 5.83 | 5.01 | 3.45 |
> | LRQ | 5.75 | 5.02 | 3.47 |
> | **SINQ** | **5.60*** | **4.96** | **3.42** |
>
> \* Also in the current version of the paper (Table 11).
>
> OmniQuant, FlexRound and LRQ numbers from Table 7 of the LRQ paper ([4] https://arxiv.org/pdf/2407.11534)
>
> EfficientQAT instead is a QAT method (Quantization-Aware Training), so it is not classified as PTQ technique (https://arxiv.org/pdf/2407.11062).
>
> To provide a comparison also to this method we used the official EfficientQAT code (https://github.com/OpenGVLab/EfficientQAT) on Llama-2-7B. The retraining time (alone, without considering the quantization) for this model is about ~4.3h while for SINQ, on the same hardware with 1xGPU, 11 seconds are sufficient to get the quantized model. The quality of EfficientQAT, given the retraining, is very high, but SINQ seems to close the gap on more OOD data (e.g. coding).
>
>
> | Method        | WikiText-2 (ppl)  | Python* [ppl]↓    |
> |:--------------|------------------:|------------------:|
> | FP16          | 5.47              | 2.50              |
> | SINQ          | 5.90              | 2.74              |
> | EfficientQAT  | 5.76              | 2.72              |
> * data codeparrot/codeparrot-clean-valid (available in HF)
>
> This is an unfair comparison for SINQ since the QAT model has been retrained + quantized using the calibrated method GPTQ. We used ChenMnZ/Llama-2-7b-EfficientQAT-w4g128-GPTQ available on Hugging Face.
>
>
> 3. "In Table 13, the authors show the experimental results of MoE, which is impressive. [...] it would be required to measure AIME24 and/or AIME25 scores in order to highlight that SINQ is also effective for MoE models."
>
> We agree with the reviewer that evaluating on reasoning tasks also MoE models is important. We chose Qwen3-235B-A22B (with thinking mode) since DeepSeek-V2.5-236B, while using MoE, is not trained as a reasoning model. Results show performance similar to the one reported in Table 7 for a dense model.
>
> | Method | AIME24 Avg@3 | AIME25 Avg@3 |
> |:-------|-------------:|-------------:|
> | FP16   | 78.90        | 75.60        |
> | *Calibration-free* |  |  |
> | RTN    | 64.40        | 57.80        |
> | **SINQ** | **73.30**  | **70.00**    |
> | *Calibrated* |  |  |
> | AWQ    | 74.40        | 67.80        |
> | **ASINQ** | **76.70** | **70.00**    |
>
> Budget: 16k tokens

---

> > ### Author Rebuttal · Reviewer_cLRS · 2026-04-01
> >
> > Thank you for the detailed response. As all of my concerns are addressed, I have increased my score accordingly.

---

> > > ### Author Response · Authors · 2026-04-07
> > >
> > > Thank you for your time and your helpful feedback.

---

### Official Review · Reviewer_pDGF · 2026-03-22

**Soundness:** 3
**Presentation:** 3
**Significance:** 3
**Originality:** 3
**Overall Recommendation:** 5
**Confidence:** 2

**Summary:**

This paper introduces SINQ, a calibration-free post-training quantization framework designed to mitigate performance degradation in LLMs with low-bit quantization. Recognizing that weight outliers often collapse quantization scales, SINQ implements a dual-axis scaling mechanism coupled with a fast Sinkhorn–Knopp-style algorithm. This approach iteratively balances row and column variances to approximate pseudo-activation awareness without relying on external calibration data. Being architecture-agnostic and layer-independent, SINQ incurs negligible computational overhead. Extensive evaluations on models demonstrate that SINQ significantly outperforms traditional calibration-free methods. Furthermore, SINQ can be combined with calibration and non-uniform quantization schemes for further improvements.

**Compliance With Llm Reviewing Policy:**

Affirmed.

**Final Justification:**

Based on the rebuttal, my concerns about model generality, benchmarks, and 2‑bit/W4A4 are fully resolved. The authors provided strong additional evidence. I have raised my score and recommend acceptance.

**Key Questions For Authors:**

Could the authors provide experimental results for 2-bit quantization to demonstrate the scalability and robustness of the SINQ under extreme bit-width constraints? In addition to PPL, could the authors include evaluations on downstream zero-shot tasks (e.g., MMLU) for the 2-bit setting?

**Limitations:**

Although SINQ demonstrates strong performance in 3/4-bit weight-only quantization, the authors do not discuss its applicability to the more challenging 2-bit ultra-low precision scenario or the weight-activation joint quantization (W4A4) setting, which to some extent limits the reference value of this method in pursuit of extreme compression and hardware acceleration solutions.

**Strengths And Weaknesses:**

Strengths:
1. In many real-world scenarios (e.g., privately fine-tuned models), obtaining high-quality calibration data is challenging. SINQ demonstrates that it is possible to achieve high accuracy using only the mathematical structure of the weights themselves, without relying on any calibration data.
2. SINQ can be combined with NF4 or integrated with calibration-data-dependent methods such as AWQ. This compatibility makes it easy to incorporate into existing quantization toolkits.

Weaknesses:
1. The core assumption of SINQ is that “the standard deviation of weight columns is inversely proportional to the magnitude of activations.” Although the authors validated the method on Qwen, the Llama series, and DeepSeek-V2.5-236B, whether this assumption holds across a broader range of model families still requires further experimental verification.
2. The evaluation is largely centered on PPL. For downstream tasks, only three benchmarks (HellaSwag, PIQA, and MMLU) are reported. A more comprehensive set of downstream benchmarks would help better assess the method’s practical effectiveness and generalizability.

---

> ### Author Rebuttal · Authors · 2026-03-31
>
> Thank you for your helpful feedback.
>
> ## Weaknesses
> 1. "The core assumption of SINQ is that 'the standard deviation of weight columns is inversely proportional to the magnitude of activations' [...] whether this assumption holds across a broader range of model families still requires further experimental verification."
>
> We will add the following corroborating results to the paper. In the table below, we report the average R^2 between mean absolute input and reciprocal column-wise standard deviation for further LLMs, SSMs, Hybrid language models, vision models and protein models. For all these models we observe a similar relationship (full table provided in response to Reviewer WMPW). XW-R^2 is the R^2 between mean absolute x and the weight standard deviation, shuffled... is the same with shuffled weight std. dev. and xx-R^2 is the same but measured on activations of different input distributions (as a plausible upper bound).
>
> | Model          | Type       | XW-R^2 | shuffled-Wstd-R^2 | xx-R^2 |
> |:---------------|:-----------|-------:|------------------:|-------:|
> | Qwen3-1.7B | LLM | 0.1825 | 0.0003 | 0.6191 |
> | Llama-2-13B    | LLM        | 0.2058 | 0.0002            | 0.5434 |
> | Mamba-2.8B | SSM | 0.1989 | 0.0021 | 0.6054 |
> | FalconH1-3B    | Hybrid-LLM | 0.1979 | 0.0003            | 0.5445 |
> | SwinV2-B       | Vision     | 0.1908 | 0.0027            | 0.8034 |
> | ESM2-35M       | Protein    | 0.3894 | 0.0124            | 0.9217 |
>
> 2. "The evaluation is largely centered on PPL. For downstream tasks, only three benchmarks (HellaSwag, PIQA, and MMLU) are reported. A more comprehensive set of downstream benchmarks would help [...]"
>
> We agree that there are cases where quantized LLMs do not generalize well, especially to reasoning tasks. Unfortunately, benchmarking on these is substantially more computationally intensive, and in practice we can only compare against methods that provide optimized kernels compatible with fast serving frameworks (e.g., vLLM or SGLang).
>
> We will add more results on GSM8k, IFEval and AIME24/25 to the paper. Also in this setting SINQ performs well. Due to the limited number of characters per review, these results are reported in our response to Reviewer cLRS.
>
> ## Questions
> 1. "Although SINQ demonstrates strong performance in 3/4-bit weight-only quantization, the authors do not discuss its applicability to the more challenging 2-bit ultra-low precision scenario or the weight-activation joint quantization (W4A4) setting [...]"
>
> To the best of our knowledge comparable methods do not implement 2-bit quantization and our key baselines do not offer 2-bit variants (HQQ, AWQ, BNB, ...). For this reason we did not report results on this.
>
> For uncalibrated PTQ methods, 2 bit weight quantization is still a very challenging scenario. QuaRot reports in the Appendix in Table 7 A16W2 results for QuaRot-RTN for Llama models and in every setting ppl is reported as Inf.
>
> However, we find that with SINQ uncalibrated 2-bit quantization does become feasible to some extent. We take a 3-bit Qwen3-1.7B (non-thinking) and progressively select layers (based on kurtosis, lowest-first) to quantize to 2-bits (note that this is far from an optimal strategy, just a prelimiary result). We see that even with this simple method on a small and hard-to-quantize model quantizing some layers very aggressively without any calibration, fine-tuning or quantization-aware training leads to graceful degradation, not catastrophic failure.
>
> ### Results Table (Wikitext-2 PPL)
>
> fp16: 16.67
>
> | `fraction_2bit` | SINQ     | RTN       |
> |:----------------|---------:|----------:|
> | 0               | 22.39    | 32.43     |
> | 1/32            | 23.58    | 35.00     |
> | 1/16            | 25.57    | 50.73     |
> | 1/8             | 36.66    | 141.63    |
> | 1/4             | 115.03   | 225.43    |
> | 1/2             | 510.86   | 3509.79   |
> | 1               | 30574.16 | 210237.47 |
>
> ### Results Table (MMLU Accuracy, 0-shot)
>
> fp16: 0.5624
>
> | `fraction_2bit` | SINQ   | RTN    |
> |:----------------|-------:|-------:|
> | 0               | 0.4477 | 0.3334 |
> | 1/32            | 0.4162 | 0.3269 |
> | 1/16            | 0.3915 | 0.2793 |
> | 1/8             | 0.3047 | 0.2676 |
> | 1/4             | 0.2671 | 0.2671 |
> | 1/2             | 0.2543 | 0.2602 |
> | 1               | 0.2496 | 0.2464 |
>
>
> We did not report results for W4A4 since our method focuses specifically on weight quantization, but the technique is fully compatible with activation quantization. We report here a simple example:
>
> | Model        | Bits | Method | Wikitext [ppl]↓ |
> |:-------------|:-----|:-------|----------------:|
> | Qwen-3 1.7B  | W4A4 | RTN    | 35.63           |
> | Qwen-3 1.7B  | W4A4 | SINQ   | 30.76           |
> | Qwen-3 14B   | W4A4 | RTN    | 10.55           |
> | Qwen-3 14B   | W4A4 | SINQ   | 10.44           |
> | Qwen-3 32B   | W4A4 | RTN    | 9.65            |
> | Qwen-3 32B   | W4A4 | SINQ   | 9.53            |

---

> > ### Author Rebuttal · Reviewer_pDGF · 2026-04-03
> >
> > The authors have addressed my concerns, so I will raise my score accordingly.

---

> > > ### Author Response · Authors · 2026-04-07
> > >
> > > Dear Reviewer, thank you for noting that all concerns have been resolved and for your intention to raise the score. Right now, the numerical score in the system remains a 4. We'd be grateful if you were to update it as intended. Thanks again for your time and all your feedback.

---

### Decision · Program_Chairs · 2026-04-30

**Decision:**

Accept (regular)

**Comment:**

This paper presents a weight-only post-training quantization method without need of calibration data, SINQ, for LLMs. The core contribution is a dual-axis scaling mechanism coupled with a fast Sinkhorn–Knopp-style algorithm to normalize per-row and per-column variances. The authors conducted experiments on multiple generation/reasoning tasks with different LLM architectures/scales to show the efficacy of SINQ.

The paper was initially/finally scored (4,3,4,5)/(5,5,4,5) by four reviewers, who mostly recognized the motivation, the idea and the empirical results, and raised several concerns about 1) whether the core assumption “the standard deviation of weight columns is inversely proportional to the magnitude of activations” and the theoretical justification would be valid to different LLMs; 2) more through experimental comparison on additional reasoning tasks; 3) how about the performance under ultra low-bit quantization settings, e.g., 2-bit; 4) some unclear points in the Method section.

The authors provided detailed responses to address these concerns. Finally, the weakly positive Reviewer L3dA did not provide any feedback and thus this review is given less weight, and the other three reviewers (pDGF, cLRS, WMPW) were satisfied with the rebuttal and consistently gave the positive score Accept. The AC read the paper, the reviews, the rebuttal and the reviewers' feedback. I agree with reviewers that this paper has an insightful motivation, good novelty and is technical sound and shows good performance, and thus recommend an “Accept”. The authors are encouraged to include additional experiments, discussions and clarifications in the final version of paper.